# Compressing Neural Networks: Towards Determining the Optimal Layer-wise Decomposition

**Lucas Liebenwein**[*]
MIT CSAIL
lucas@csail.mit.edu

**Alaa Maalouf**[*]
University of Haifa
alaamalouf12@gmail.com

**Oren Gal**
University of Haifa
orengal@alumni.technion.ac.il

**Dan Feldman**
University of Haifa
dannyf.post@gmail.com

**Daniela Rus**
MIT CSAIL
rus@csail.mit.edu

## Abstract

We present a novel global compression framework for deep neural networks that automatically analyzes each layer to identify the optimal per-layer compression ratio, while simultaneously achieving the desired overall compression. Our algorithm hinges on the idea of compressing each convolutional (or fully-connected) layer by slicing its channels into multiple groups and decomposing each group via low-rank decomposition. At the core of our algorithm is the derivation of layer-wise error bounds from the Eckart–Young–Mirsky theorem. We then leverage these bounds to frame the compression problem as an optimization problem where we wish to minimize the maximum compression error across layers and propose an efficient algorithm towards a solution. Our experiments indicate that our method outperforms existing low-rank compression approaches across a wide range of networks and data sets. We believe that our results open up new avenues for future research into the global performance-size trade-offs of modern neural networks.

## 1 Introduction

Neural network compression entails taking an existing model and reducing its computational and memory footprint in order to enable the deployment of large-scale networks in resource-constrained environments. Beyond inference time efficiency, compression can yield novel insights into the design (Liu et al., 2019b), training (Liebenwein et al., 2021a,b), and theoretical properties (Arora et al., 2018) of neural networks.

Among existing compression techniques – which include quantization (Wu et al., 2016), distillation (Hinton et al., 2015), and pruning (Han et al., 2015) – *low-rank compression* aims at decomposing a layer's weight tensor into a tuple of smaller low-rank tensors. Such compression techniques may build upon the rich literature on low-rank decomposition and its numerous applications outside deep learning such as dimensionality reduction (Laparra et al., 2015) or spec-

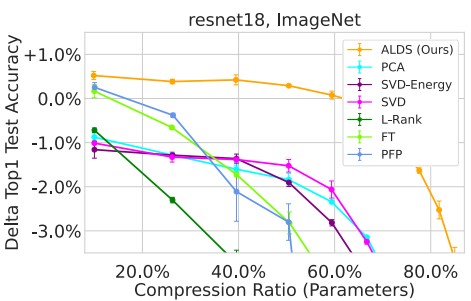

Figure 1: **ALDS**, *Automatic Layer-wise Decomposition Selector*, can compress up to 60% of parameters on a ResNet18 (ImageNet), 3x more compared to baselines. Detailed results are described in Section 3.

---

[*]denotes authors with equal contributions. Code: https://github.com/lucaslie/torchprune

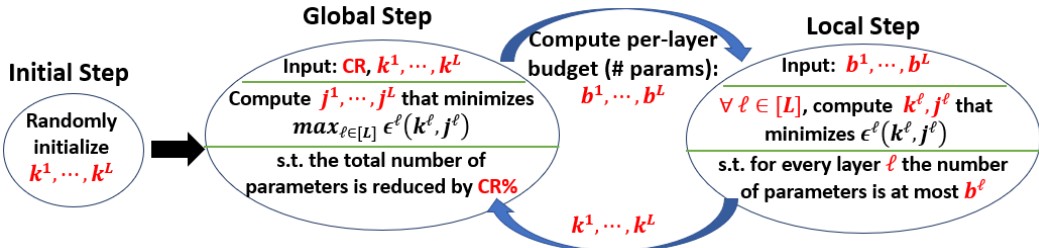

Figure 2: **ALDS Overview**. The framework consists of a global and local step, see Section 2.

tral clustering (Peng et al., 2015). Moreover, low-rank compression can be readily implemented in any machine learning framework by replacing the existing layer with a set of smaller layers without the need for, e.g., sparse linear algebra support.

Within deep learning, we encounter two related, yet distinct challenges when applying low-rank compression. On the one hand, each layer should be efficiently decomposed (the "local step") and, on the other hand, we need to balance the amount of compression in each layer in order to achieve a desired overall compression ratio with minimal loss in the predictive power of the network (the "global step"). While the "local step", i.e., designing the most efficient layer-wise decomposition method, has traditionally received lots of attention (Denton et al., 2014; Garipov et al., 2016; Jaderberg et al., 2014; Kim et al., 2015b; Lebedev et al., 2015; Novikov et al., 2015), the "global step" has only recently been the focus of attention in research, e.g., see the recent works of Alvarez and Salzmann (2017); Idelbayev and Carreira-Perpinán (2020); Xu et al. (2020).

In this paper, we set out to design a framework that simultaneously accounts for both the local and global step. Our proposed solution, termed *Automatic Layer-wise Decomposition Selector* (ALDS), addresses this challenge by iteratively optimizing for each layer's decomposition method (local step) and the low-rank compression itself while accounting for the maximum error incurred across layers (global step). In Figure 1, we show how ALDS outperforms existing approaches on the common ResNet18 (ImageNet) benchmark ($60\%$ compression compared to $\sim 20\%$ for baselines).

**Efficient layer-wise decomposition.** Our framework relies on a straightforward SVD-based decomposition of each layer. Inspired by Denton et al. (2014); Idelbayev and Carreira-Perpinán (2020); Jaderberg et al. (2014) and others, we decompose each layer by first folding the weight tensor into a matrix before applying SVD and encoding the resulting pair of matrices as two separate layers.

**Enhanced decomposition via multiple subsets.** A natural generalization of low-rank decomposition methods entails splitting the matrix into multiple subsets (subspaces) before compressing each subset individually. In the context of deep learning, this was investigated before for individual layers (Denton et al., 2014), including embedding layers (Chen et al., 2018; Maalouf et al., 2021). We take this idea further and incorporate it into our layer-wise decomposition method as additional hyperparameter in terms of the number of subsets. Thus, our local step, i.e., the layer-wise decomposition, constitutes of choosing the number of subsets ($k^\ell$) for each layer and the rank ($j^\ell$).

**Towards a global solution for low-rank compression.** We can describe the optimal solution for low-rank compression as the set of hyperparameters (number of subspaces $k^\ell$ and rank $j^\ell$ for each layer in our case) that minimizes the drop in accuracy of the compressed network. While finding the globally optimal solution is NP-complete, we propose ALDS as an efficiently solvable alternative that enables us to search for a locally optimal solution in terms of the maximum relative error incurred across layers. To this end, we derive spectral norm bounds based on the Eckhart-Young-Mirsky Theorem for our layer-wise decomposition method to describe the trade-off between the layer compression and the incurred error. Leveraging our bounds we can then efficiently optimize over the set of possible per-layer decompositions. An overview of ALDS is shown in Figure 2.

## 2 Method

In this section, we introduce our compression framework consisting of a layer-wise decomposition method (Section 2.1), a global selection mechanism to simultaneously compress all layers of a network (Section 2.2), and an optimization procedure (ALDS) to solve the selection problem (Section 2.3).

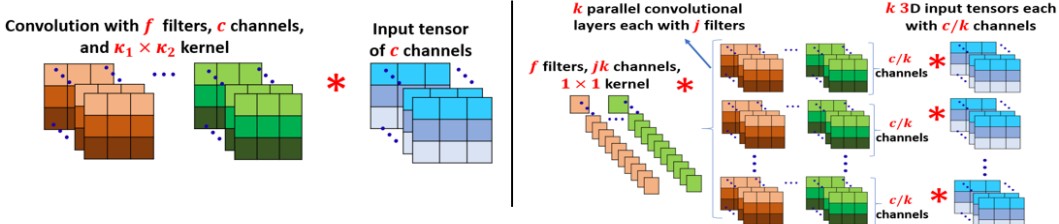

Figure 3: **Left: 2D convolution. right: decomposition used for ALDS.** For a $f \times c \times \kappa_1 \times \kappa_2$ convolution with $f$ filters, $c$ channels, and $\kappa_1 \times \kappa_2$ kernel, our per-layer decomposition consists: (1) $k$ parallel $j \times c/k \times \kappa_1 \times \kappa_2$ convolutions; (2) a single $f \times kj \times 1 \times 1$ convolution applied on the first layer's (stacked) output.

## 2.1 Local Layer Compression

We detail our low-rank compression scheme for convolutional layers below and note that it readily applies to fully-connected layers as well as a special case of convolutions with a $1 \times 1$ kernel.

**Compressing convolutions via SVD.** Given a convolutional layer of $f$ filters, $c$ channels, and a $\kappa_1 \times \kappa_2$ kernel we denote the corresponding weight tensor by $\mathcal{W} \in \mathbb{R}^{f \times c \times \kappa_1 \times \kappa_2}$. Following Denton et al. (2014); Idelbayev and Carreira-Perpiñán (2020); Wen et al. (2017) and others, we can then interpret the layer as a linear layer of shape $f \times c\kappa_1\kappa_2$ and the corresponding rank $j$-approximation as two subsequent linear layers of shape $f \times j$ and $j \times c\kappa_1\kappa_2$. Mapped back to convolutions, this corresponds to a $j \times c \times \kappa_1 \times \kappa_2$ convolution followed by a $f \times j \times 1 \times 1$ convolution.

**Multiple subspaces.** Following the intuition outlined in Section 1 we propose to cluster the columns of the layer's weight matrix into $k \geq 2$ separate subspaces before applying SVD to each subset. To this end, we may consider any clustering method, such as k-means or projective clustering (Chen et al., 2018; Maalouf et al., 2021). However, such methods require expensive approximation algorithms which would limit our ability to incorporate them into an optimization-based compression framework as outlined in Section 2.2. In addition, arbitrary clustering may require re-shuffling the input tensors which could lead to significant slow-downs during inference. We instead opted for a simple clustering method, namely *channel slicing*, where we simply divide the $c$ input channels of the layer into $k$ subsets each containing at most $\lceil c/k \rceil$ consecutive input channels. Unlike other methods, channel slicing is efficiently implementable, e.g., as grouped convolutions in PyTorch (Paszke et al., 2017) and ensures practical speed-ups subsequent to compressing the network.

**Overview of per-layer decomposition.** In summary, for given integers $j, k \geq 1$ and a 4D tensor $\mathcal{W} \in \mathbb{R}^{f \times c \times \kappa_1 \times \kappa_2}$ representing a convolution the per-layer compression method proceeds as follows:

1. PARTITION the channels of the convolutional layer into $k$ subsets, where each subset has at most $\lceil c/k \rceil$ consecutive channels, resulting in $k$ convolutional tensors $\{\mathcal{W}_i\}_{i=1}^k$ where $\mathcal{W}_i \in \mathbb{R}^{f \times c_i \times \kappa_1 \times \kappa_2}$, and $\sum_{i=1}^k c_i = c$.

2. DECOMPOSE each tensor $\mathcal{W}_i$, $i \in [k]$, by building the corresponding weight matrix $W_i \in \mathbb{R}^{f \times c_i \kappa_1 \kappa_2}$, c.f. Figure 3, computing its $j$-rank approximation, and factoring it into a pair of smaller matrices $U_i$ of $f$ rows and $j$ columns and $V_i$ of $j$ rows and $c_i \kappa_1 \kappa_2$ columns.

3. REPLACE the original layer in the network by 2 layers. The first consists of $k$ parallel convolutions, where the $i^{\text{th}}$ parallel layer, $i \in [k]$, is described by the tensor $\mathcal{V}_i \in \mathbb{R}^{j \times c_i \times \kappa_1 \times \kappa_2}$ which can be constructed from the matrix $V_i$ ($j$ filters, $c_i$ channels, $\kappa_1 \times \kappa_2$ kernel). The second layer is constructed by reshaping each matrix $U_i$, $i \in [k]$, to obtain the tensor $\mathcal{U}_i \in \mathbb{R}^{f \times j \times 1 \times 1}$, and then channel stacking all $k$ tensors $\mathcal{U}_1, \cdots, \mathcal{U}_k$ to get a single tensor of shape $f \times kj \times 1 \times 1$.

The decomposed layer is depicted in Figure 3. The resulting layer pair has $jc\kappa_1\kappa_2$ and $jfk$ parameters, respectively, which implies a parameter reduction from $fc\kappa_1\kappa_2$ to $j(fk + c\kappa_1\kappa_2)$.

## 2.2 Global Network Compression

In the previous section, we introduced our layer compression scheme. We note that in practice we usually want to compress an entire network consisting of $L$ layers up to a pre-specified relative reduction in parameters ("compression ratio" or **CR**). However, it is generally unclear how much

each layer $\ell \in [L]$ should be compressed in order to achieve the desired **CR** while incurring a minimal increase in loss. Unfortunately, this optimization problem is NP-complete as we would have to check every combination of layer compression resulting in the desired **CR** in order to optimally compress each layer. On the other hand, simple heuristics, e.g., constant per-layer compression ratios, may lead to sub-optimal results, see Section 3. To this end, we propose an efficiently solvable global compression framework based on minimizing the maximum relative error incurred across layers. We describe each component of our optimization procedure in greater detail below.

**The layer-wise relative error as proxy for the overall loss.**   Since the true cost (the additional loss incurred after compression) would result in an NP-complete problem, we replace the true cost by a more efficient proxy. Specifically, we consider the maximum relative error $\varepsilon := \max_{\ell \in [L]} \varepsilon^\ell$ across layers, where $\varepsilon^\ell$ denotes the theoretical maximum relative error in the $\ell^{\text{th}}$ layer as described in Theorem 1 below. We choose to minimize this particular cost because: (i) minimizing the maximum relative error ensures that no layer incurs an unreasonably large error that might otherwise get propagated or amplified; (ii) relying on a relative instead of an absolute error notion is preferred as scaling between layers may arbitrarily change, e.g., due to batch normalization, and thus the absolute scale of layer errors may not be indicative of the increase in loss; and (iii) the per-layer relative error has been shown to be intrinsically linked to the theoretical compression error, e.g., see the works of Arora et al. (2018) and Baykal et al. (2019a) thus representing a natural proxy for the cost.

**Definition of per-layer relative error.**   Let $\mathcal{W}^\ell \in \mathbb{R}^{f^\ell \times c^\ell \times \kappa_1^\ell \times \kappa_2^\ell}$ and $W^\ell \in \mathbb{R}^{f^\ell \times c^\ell \kappa_1^\ell \kappa_2^\ell}$ denote the weight tensor and corresponding folded matrix of layer $\ell$, respectively. The per-layer relative error $\varepsilon^\ell$ is hereby defined as the relative difference in the operator norm between the matrix $\hat{W}^\ell$ (that corresponds to the compressed weight tensor $\hat{\mathcal{W}}^\ell$) and the original weight matrix $W^\ell$ in layer $\ell$, i.e,.

$$\varepsilon^\ell := \|\hat{W}^\ell - W^\ell\| / \|W^\ell\|. \tag{1}$$

Note that while in practice our method decomposes the original layer into a set of separate layers (see Section 2.1), for the purpose of deriving the resulting error we re-compose the compressed layers into the overall matrix operator $\hat{W}^\ell$, i.e., $\hat{W}^\ell = [U_1^\ell V_1^\ell \cdots U_{k^\ell}^\ell V_{k^\ell}^\ell]$, where $U_i^\ell V_i^\ell$ is the factorization of the $i$th cluster (set of columns) in the $\ell$th layer, for every $\ell \in [L]$ and $i \in [k^\ell]$, see supplementary material for more details. We note that the operator norm $\| \cdot \|$ for a convolutional layer thus signifies the maximum relative error incurred for an individual output patch ("pixel") across all output channels.

**Derivation of relative error bounds.**   We now derive an error bound that enables us to describe the per-layer relative error in terms of the compression hyperparameters $j^\ell$ and $k^\ell$, i.e., $\varepsilon^\ell = \varepsilon^\ell(k^\ell, j^\ell)$. This will prove useful later on as we have to repeatedly query the relative error in our optimization procedure. The error bound is described in the following.

**Theorem 1.** *Given a layer matrix $W^\ell$ and the corresponding low-rank approximation $\hat{W}^\ell$, the relative error $\varepsilon^\ell := \|\hat{W}^\ell - W^\ell\| / \|W^\ell\|$ is bounded by*

$$\varepsilon^\ell \le \sqrt{k}/\alpha_1 \cdot \max_{i \in [k]} \alpha_{i,j+1}, \tag{2}$$

*where $\alpha_{i,j+1}$ is the $j+1$ largest singular value of the matrix $W_i^\ell$, for every $i \in [k]$, and $\alpha_1 = \|W^\ell\|$ is the largest singular value of $W^\ell$.*

*Proof.* First, we recall the matrices $W_1^\ell, \cdots, W_k^\ell$ and we denote the SVD factorization for each of them by: $W_i^\ell = \tilde{U}_i^\ell \tilde{\Sigma}_i^\ell \tilde{V}_i^\ell$. Now, observe that for every $i \in [k]$, the matrix $\hat{W}_i^\ell$ is the $j$-rank approximation of $W_i^\ell$. Hence, the SVD factorization of $\hat{W}_i^\ell$ can be written as $\hat{W}_i^\ell = \tilde{U}_i^\ell \hat{\Sigma}_i^\ell \tilde{V}_i^{\ell T}$, where $\hat{\Sigma}_i^\ell \in \mathbb{R}^{f \times d}$ is a diagonal matrix such that its first $j$-diagonal entries are equal to the first $j$-entries on the diagonal of $\tilde{\Sigma}_i^\ell$, and the rest are zeros. Hence,

$$W^\ell - \hat{W}^\ell = [W_1^\ell - \hat{W}_1^\ell, \cdots, W_k^\ell - \hat{W}_k^\ell] = [\tilde{U}_1^\ell (\tilde{\Sigma}_1^\ell - \hat{\Sigma}_1^\ell) \tilde{V}_1^\ell, \cdots, \tilde{U}_k^\ell (\tilde{\Sigma}_k^\ell - \hat{\Sigma}_k^\ell) \tilde{V}_k^\ell]$$
$$= [\tilde{U}_1^\ell \cdots \tilde{U}_k^\ell] \operatorname{diag} \left( (\tilde{\Sigma}_1^\ell - \hat{\Sigma}_1^\ell) \tilde{V}_1^\ell, \ldots, (\tilde{\Sigma}_k^\ell - \hat{\Sigma}_k^\ell) \tilde{V}_k^\ell \right). \tag{3}$$

By (3) and by the triangle inequality, we have that

$$\left\| W^\ell - \hat{W}^\ell \right\| \le \left\| \left[ \tilde{U}_1^\ell \cdots \tilde{U}_k^\ell \right] \right\| \left\| \operatorname{diag} \left( (\tilde{\Sigma}_1^\ell - \hat{\Sigma}_1^\ell) \tilde{V}_1^\ell, \ldots, (\tilde{\Sigma}_k^\ell - \hat{\Sigma}_k^\ell) \tilde{V}_k^\ell \right) \right\|. \tag{4}$$

Now, we observe that

$$\left\| \left[ \tilde{U}_1^\ell \cdots \tilde{U}_k^\ell \right] \right\|^2 = \left\| \left[ \tilde{U}_1^\ell \cdots \tilde{U}_k^\ell \right] \left[ \tilde{U}_1^\ell \cdots \tilde{U}_k^\ell \right]^T \right\| = \| \mathrm{diag}(k, \ldots, k) \| = k. \tag{5}$$

Finally, we show that

$$\left\| \mathrm{diag} \left( (\tilde{\Sigma}_1^\ell - \hat{\Sigma}_1^\ell) \tilde{V}_1^\ell, \ldots, (\tilde{\Sigma}_k^\ell - \hat{\Sigma}_k^\ell) \tilde{V}_k^\ell \right) \right\| = \max_{i \in [k]} \left\| (\tilde{\Sigma}_i^\ell - \hat{\Sigma}_i^\ell) \tilde{V}_i^\ell \right\| \tag{6}$$

$$= \max_{i \in [k]} \left\| (\tilde{\Sigma}_i^\ell - \hat{\Sigma}_i^\ell) \right\| = \max_{i \in [k]} \alpha_{i,j+1}, \tag{7}$$

where the second equality holds since the columns of $V$ are orthogonal and the last equality holds according to the Eckhart-Young-Mirsky Theorem (Theorem 2.4.8 of Golub and Van Loan (2013)). Plugging (7) and (5) into (4) concludes the proof. □

**Resulting network size.** Let $\theta = \{\mathcal{W}^\ell\}_{\ell=1}^L$ denote the set of weights for the $L$ layers and note that the number of parameters in layer $\ell$ is given by $|\mathcal{W}^\ell| = f^\ell c^\ell \kappa_1^\ell \kappa_2^\ell$ and $|\theta| = \sum_{\ell \in [L]} |\mathcal{W}^\ell|$. Moreover, note that $|\hat{\mathcal{W}}^\ell| = j^\ell (k^\ell f^\ell + c^\ell \kappa_1^\ell \kappa_2^\ell)$ if decomposed, $\hat{\theta} = \{\hat{\mathcal{W}}^\ell\}_{\ell=1}^L$, and $|\hat{\theta}| = \sum_{\ell \in [L]} |\hat{\mathcal{W}}^\ell|$. The overall compression ratio is thus given by $1 - |\hat{\theta}|/|\theta|$ where we neglected other parameters for ease of exposition. Observe that the layer budget $|\hat{\mathcal{W}}^\ell|$ is fully determined by $k^\ell, j^\ell$ just like the error bound.

**Global Network Compression.** Putting everything together we obtain the following formulation for the optimal per-layer budget:

$$\varepsilon_{opt} = \min_{\{j^\ell, k^\ell\}_{\ell=1}^L} \quad \max_{\ell \in [L]} \varepsilon^\ell(k^\ell, j^\ell) \tag{8}$$
$$\text{subject to} \quad 1 - |\hat{\theta}(k^1, j^1, \ldots, k^L, j^L)|/|\theta| \leq \mathbf{CR},$$

where $\mathbf{CR}$ denotes the desired overall compression ratio. Thus optimally allocating a per-layer budget entails finding the optimal number of subspaces $k^\ell$ and ranks $j^\ell$ for each layer constrained by the desired overall compression ratio $\mathbf{CR}$.

## 2.3 Automatic Layer-wise Decomposition Selector (ALDS)

We propose to solve (8) by iteratively optimizing $k^1, \ldots, k^L$ and $j^1, \ldots, j^L$ until convergence akin of an EM-like algorithm as shown in Algorithm 1 and Figure 2.

Specifically, for a given set of weights $\theta$ and desired compression ratio $\mathbf{CR}$ we first randomly initialize the number of subspaces $k^1, \ldots, k^L$ for each layer (Line 2). Based on given values for each $k^\ell$ we then solve for the optimal ranks $j^1, \ldots, j^L$ such that the overall compression ratio is satisfied (Line 4). Note that the maximum error $\varepsilon$ is minimized if all errors are equal. Thus solving for the

---

**Algorithm 1** ALDS($\theta$, $\mathbf{CR}$, $n_{\text{seed}}$)

**Input:** $\theta$: network parameters; $\mathbf{CR}$: overall compression ratio; $n_{\text{seed}}$: number of random seeds to initialize
**Output:** $k^1, \ldots, k^L$: number of subspaces for each layer; $j_1, \ldots, j^L$: desired rank per subspace for each layer

1: **for** $i \in [n_{\text{seed}}]$ **do**
2:     $k^1, \ldots, k^L \leftarrow$ RANDOMINIT()
3:     **while** not converged **do**
4:         $j^1, \ldots, j^L \leftarrow$ OPTIMALRANKS($\mathbf{CR}, k^1, \ldots, k^L$)    ▷ Global step: choose s.t. $\varepsilon^1 = \ldots = \varepsilon^L$
5:         **for** $\ell \in [L]$ **do**
6:             $b^\ell \leftarrow j^\ell(k^\ell f^\ell + c^\ell \kappa_1^\ell \kappa_2^\ell)$    ▷ resulting layer budget
7:             $k^\ell \leftarrow$ OPTIMALSUBSPACES($b^\ell$)    ▷ Local step: minimize error bound for a given layer budget
8:         **end for**
9:     **end while**
10:    $\varepsilon_i =$ RECORDERROR($k^1, \ldots, k^L, j^1, \ldots, j^L$)
11: **end for**
12: **return** $k^1, \ldots, k^L, j^1, \ldots, j^L$ from $i_{\text{best}} = \mathrm{argmin}_i \varepsilon_i$

---

ranks in Line 4 entails guessing a value for $\varepsilon$, computing the resulting network size, and repeating the process until the desired **CR** is satisfied, e.g. via binary search.

Subsequently, we re-assign the number of subspaces $k^\ell$ for each layer by iterating through the finite set of possible values for $k^\ell$ (Line 7) and choosing the one that minimizes the relative error for the current layer budget $b^\ell$ (computed in Line 6). Note that we can efficiently approximate the relative error by leveraging Theorem 1. We then iteratively repeat both steps until convergence (Lines 3-8). To improve the quality of the local optimum we initialize the procedure with multiple random seeds (Lines 1-11) and pick the allocation with the lowest error (Line 12).

We note that we make repeated calls to our decomposition subroutine (i.e. SVD; Lines 4, 7) highlighting the necessity for it to be efficient and cheap to evaluate. Moreover, we can further reduce the computational complexity by leveraging Theorem 1 as mentioned above.

Additional details pertaining to ALDS are provided in the supplementary material.

**Extensions.** Here, we use SVD with multiple subspaces as per-layer compression method. However, we note that ALDS can be readily extended to any desired *set* of low-rank compression techniques. Specifically, we can replace the local step of Line 7 by a search over different methods, e.g., Tucker decomposition, PCA, or other SVD compression schemes, and return the best method for a given budget. In general, we may combine ALDS with any low-rank compression as long as we can efficiently evaluate the per-layer error of the compression scheme. In the supplementary material, we discuss some preliminary results that highlight the promising performance of such extensions.

## 3 Experiments

**Networks and datasets.** We study various standard network architectures and data sets. Particularly, we test our compression framework on ResNet20 (He et al., 2016), DenseNet22 (Huang et al., 2017), WRN16-8 (Zagoruyko and Komodakis, 2016), and VGG16 (Simonyan and Zisserman, 2015) on CIFAR10 (Torralba et al., 2008); ResNet18 (He et al., 2016), AlexNet (Krizhevsky et al., 2012), and MobileNetV2 (Sandler et al., 2018) on ImageNet (Russakovsky et al., 2015); and on Deeplab-V3 (Chen et al., 2017) with a ResNet50 backbone on Pascal VOC segmentation data (Everingham et al., 2015).

**Baselines.** We compare ALDS to a diverse set of low-rank compression techniques. Specifically, we have implemented PCA (Zhang et al., 2015b), SVD with energy-based layer allocation (SVD-Energy) following Alvarez and Salzmann (2017); Wen et al. (2017), and simple SVD with constant per-layer compression (Denton et al., 2014). Additionally, we also implemented the recent learned rank selection mechanism (L-Rank) of Idelbayev and Carreira-Perpinán (2020). Finally, we implemented two recent filter pruning methods, i.e., FT of Li et al. (2016) and PFP of Liebenwein et al. (2020), as alternative compression techniques for densely compressed networks. Additional comparisons on ImageNet are provided in Section 3.2.

**Retraining.** For our experiments, we study one-shot and iterative learning rate rewinding inspired by Renda et al. (2020) for various amounts of retraining. In particular, we consider the following unified compress-retrain pipeline across all methods:

1. TRAIN for $e$ epochs according to the standard training schedule for the respective network.
2. COMPRESS the network according to the chosen method.

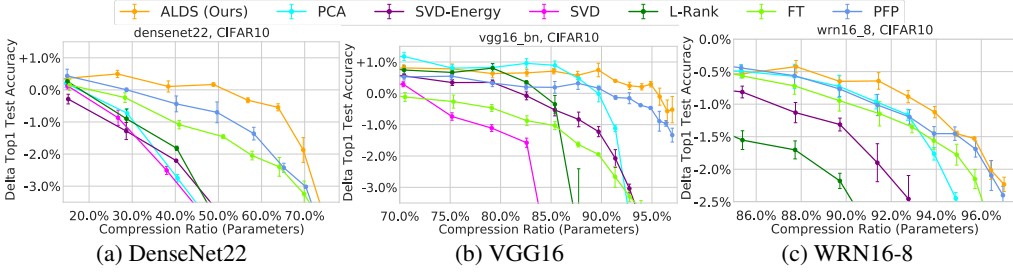

Figure 4: One-shot compress+retrain experiments on CIFAR10 with baseline comparisons.

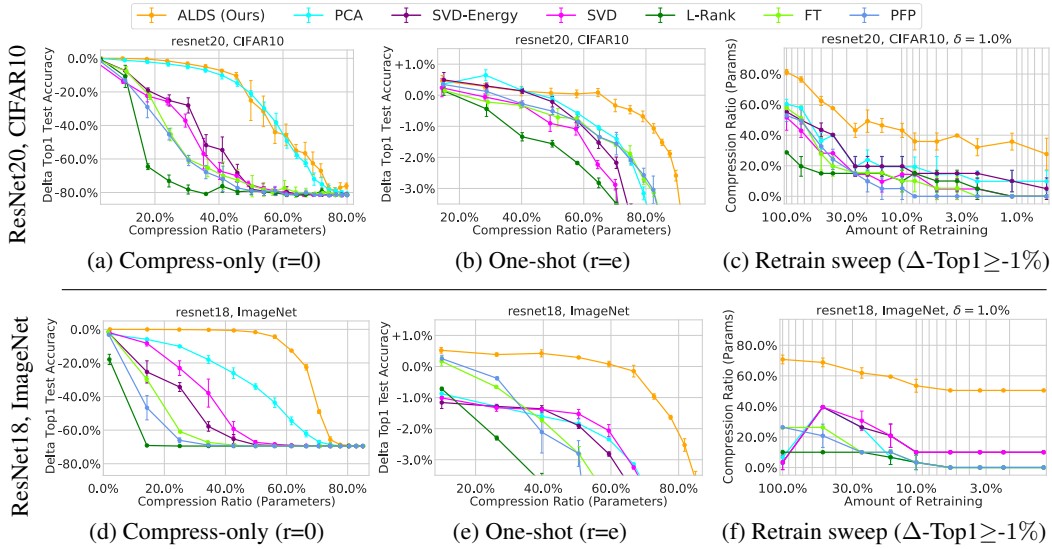

Figure 5: The size-accuracy trade-off for various compression ratios, methods, and networks. Compression was performed after training and networks were re-trained once for the indicated amount (**one-shot**). (a, b, d, e): the difference in test accuracy for fixed amounts of retraining. (c, f): the maximal compression ratio with less-than-1% accuracy drop for variable amounts of retraining.

3. RETRAIN the network for $r$ epochs using the training hyperparameters from epochs $[e - r, e]$.

4. ITERATIVELY repeat 1.-3. after projecting the decomposed layers back (optional).

**Reporting metrics.** We report Top-1, Top-5, and IoU test accuracy as applicable for the respective task. For each compressed network we also report the compression ratio, i.e., relative reduction, in terms of parameters and floating point operations denoted by CR-P and CR-F, respectively. Each experiment was repeated 3 times and we report mean and standard deviation.

## 3.1 One-shot Compression on CIFAR10, ImageNet, and VOC with Baselines

We train reference networks on CIFAR10, ImageNet, and VOC, and then compress and retrain the networks *once* with $r = e$ for various baseline comparisons and compression ratios.

**CIFAR10.** In Figure 4, we provide results for DenseNet22, VGG16, and WRN16-8 on CIFAR10. Notably, our approach is able to outperform existing baselines approaches across a wide range of tested compression ratios. Specifically, in the region where the networks incur only minimal drop in accuracy ($\Delta$-Top1$\geq-1\%$) ALDS is particularly effective.

**ResNets (CIFAR10 and ImageNet).** Moreover, we tested ALDS on ResNet20 (CIFAR10) and ResNet18 (ImageNet) as shown in Figure 5. For these experiments, we performed a grid search

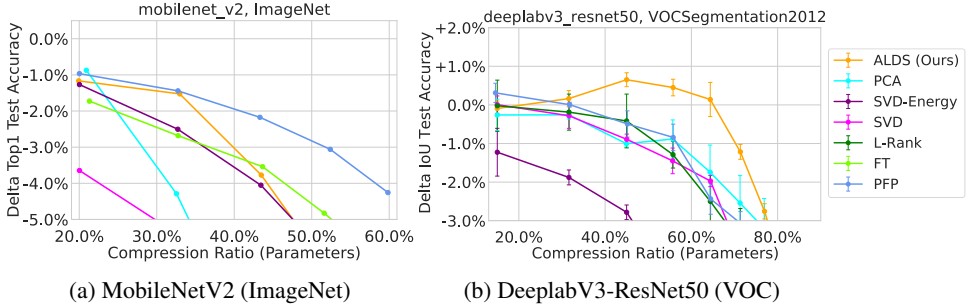

Figure 6: One-shot compress+retrain experiments on various architectures and datasets with baseline comparisons.

Table 1: Baseline results for $\Delta$-Top1$\geq-0.5\%$ for one-shot with highest CR-P and CR-F among tensor decomposition methods bolded for each network. Results coincide with Figures 4, 5, 6b.

| | Model | Metric | Tensor decomposition | | | | | Filter pruning | |
| | | | ALDS (Ours) | PCA | SVD-Energy | SVD | L-Rank | FT | PFP |
|---|---|---|---|---|---|---|---|---|---|
| CIFAR10 | ResNet20 Top1: 91.39 | $\Delta$-Top1 | -0.47 | -0.11 | -0.21 | -0.29 | -0.44 | -0.32 | -0.28 |
| | | CR-P, CR-F | **74.91, 67.86** | 49.88, 48.67 | 49.88, 49.08 | 39.81, 38.95 | 28.71, 54.89 | 39.69, 39.57 | 40.28, 30.06 |
| | VGG16 Top1: 92.78 | $\Delta$-Top1 | -0.11 | -0.02 | -0.08 | +0.29 | -0.35 | -0.47 | -0.47 |
| | | CR-P, CR-F | **95.77, 86.23** | 89.72, 85.84 | 82.57, 81.32 | 70.35, 70.13 | 85.38, 75.86 | 79.13, 78.44 | 94.87, 84.76 |
| | DenseNet22 Top1: 89.88 | $\Delta$-Top1 | -0.32 | +0.20 | -0.29 | +0.13 | +0.26 | -0.24 | -0.44 |
| | | CR-P, CR-F | **56.84, 61.98** | 14.67, 34.55 | 15.16, 19.34 | 15.00, 15.33 | 14.98, 35.21 | 28.33, 29.50 | 40.24, 43.37 |
| | WRN16-8 Top1: 89.88 | $\Delta$-Top1 | -0.42 | -0.49 | -0.41 | -0.96 | -0.45 | -0.32 | -0.44 |
| | | CR-P, CR-F | **87.77**, 79.90 | 85.33, **83.45** | 64.75, 60.94 | 40.20, 39.97 | 49.86, 58.00 | 82.33, 75.97 | 85.33, 80.68 |
| ImageNet | ResNet18 Top1: 69.62, Top5: 89.08 | $\Delta$-Top1, Top5 | -0.40, -0.05 | -0.95,-0.37 | -1.49, -0.64 | -1.75, -0.72 | -0.71, -0.23 | +0.10, +0.42 | -0.39, -0.08 |
| | | CR-P, CR-F | **66.70, 43.51** | 9.99, 12.78 | 39.56, 40.99 | 50.38, 50.37 | 10.01, 32.64 | 9.86, 11.17 | 26.35, 17.96 |
| | MobileNetV2 Top1: 71.85, Top5: 90.33 | $\Delta$-Top1, Top5 | -1.53, -0.73 | -0.87, -0.55 | -1.27, -0.57 | -3.65, -2.07 | -19.08, -13.40 | -1.73, -0.85 | -0.97, -0.40 |
| | | CR-P, CR-F | **32.97, 11.01** | 20.91, 0.26 | 20.02, 8.57 | 20.03, 31.99 | 20.00, 61.97 | 21.31, 20.23 | 20.02, 7.96 |
| VOC | DeeplabV3 IoU: 91.39 Top1: 99.34 | $\Delta$-IoU, Top1 | +0.14, -0.15 | -0.26, -0.02 | -1.88, -0.47 | -0.28, -0.18 | -0.42, -0.09 | -4.30, -0.91 | -0.49, -0.21 |
| | | CR-P, CR-F | **64.38, 64.11** | 55.68, 55.82 | 31.61, 32.27 | 31.64, 31.51 | 44.99, 45.02 | 15.00, 15.06 | 45.17, 43.93 |

over both multiple compression ratios and amounts of retraining. Here, we highlight that ALDS outperforms baseline approaches even with significantly less retraining. On Resnet 18 (ImageNet) ALDS can compress over 50% of the parameters with minimal retraining (1% retraining) and a less-than-1% accuracy drop compared to the best comparison methods (40% compression with 50% retraining).

**MobileNetV2 (ImageNet).** Next, we tested and compared ALDS on the MobileNetV2 architecture for ImageNet as shown in Figure 6a. Unlike the other networks, MobileNetV2 is a network already specifically optimized for efficient deployment and includes layer structures such as depth-wise and channel-wise convolutional operations. It is thus more challenging to find redundancies in the architecture. We find that ALDS can outperform existing tensor decomposition methods in this scenario as well.

**VOC.** Finally, we tested the same setup on a DeeplabV3 with a ResNet50 backbone trained on Pascal VOC 2012 segmentation data, see Figure 6b. We note that ALDS consistently outperforms other baselines methods in this setting as well (60% CR-P vs. 20% without accuracy drop).

**Tabular results.** Our one-shot results are again summarized in Table 1 where we report CR-P and CR-F for $\Delta$-Top1$\geq-0.5\%$. We observe that ALDS consistently improves upon prior work. We note that pruning usually takes on the order of seconds and minutes for CIFAR and ImageNet, respectively, which is usually faster than even a single training epoch.

### 3.2 ImageNet Benchmarks

Next, we test our framework on two common ImageNet benchmarks, ResNet18 and AlexNet. We follow the compress-retrain pipeline outlined in the beginning of the section and repeat it it-

Table 2: AlexNet and ResNet18 Benchmarks on ImageNet. We report Top-1, Top-5 accuracy and percentage reduction of FLOPs (CR-F). Best results with less than 0.5% accuracy drop are bolded.

| | Method | $\Delta$-Top1 | $\Delta$-Top5 | CR-F (%) |
|---|---|---|---|---|
| ResNet18, Top1, 5: 69.64%, 88.98% | ALDS (Ours) | **-0.38** | **+0.04** | **64.5** |
| | ALDS (Ours) | -1.37 | -0.56 | 76.3 |
| | MUSCO (Gusak et al., 2019) | -0.37 | -0.20 | 58.67 |
| | TRP1 (Xu et al., 2020) | -4.18 | -2.5 | 44.70 |
| | TRP1+Nu (Xu et al., 2020) | -4.25 | -2.61 | 55.15 |
| | TRP2+Nu (Xu et al., 2020) | -4.3 | -2.37 | 68.55 |
| | PCA (Zhang et al., 2015b) | -6.54 | -4.54 | 29.07 |
| | Expand (Jaderberg et al., 2014) | -6.84 | -5.26 | 50.00 |
| | PFP (Liebenwein et al., 2020) | -2.26 | -1.07 | 29.30 |
| | SoftNet (He et al., 2018) | -2.54 | -1.2 | 41.80 |
| | Median (He et al., 2019) | -1.23 | -0.5 | 41.80 |
| | Slimming (Liu et al., 2017) | -1.77 | -1.19 | 28.05 |
| | Low-cost (Dong et al., 2017) | -3.55 | -2.2 | 34.64 |
| | Gating (Hua et al., 2018) | -1.52 | -0.93 | 37.88 |
| | FT (He et al., 2017) | -3.08 | -1.75 | 41.86 |
| | DCP (Zhuang et al., 2018) | -2.19 | -1.28 | 47.08 |
| | FBS (Gao et al., 2018) | -2.44 | -1.36 | 49.49 |
| AlexNet, Top1, 5: 57.30%, 80.20% | ALDS (Ours) | -0.21 | -0.36 | 77.9 |
| | ALDS (Ours) | **-0.41** | **-0.54** | **81.4** |
| | Tucker (Kim et al., 2015a) | N/A | -1.87 | 62.40 |
| | Regularize (Tai et al., 2015) | N/A | -0.54 | 74.35 |
| | Coordinate (Wen et al., 2017) | N/A | -0.34 | 62.82 |
| | Efficient (Kim et al., 2019) | -0.7 | -0.3 | 62.40 |
| | L-Rank (Idelbayev et al., 2020) | -0.13 | -0.13 | 66.77 |
| | NISP (Yu et al., 2018) | -1.43 | N/A | 67.94 |
| | OICSR (Li et al., 2019a) | -0.47 | N/A | 53.70 |
| | Oracle (Ding et al., 2019) | -1.13 | -0.67 | 31.97 |

eratively to obtain higher compression ratios. Specifically, after retraining and before the next compression step we project the decomposed layers back to the original layer. This way, we avoid recursing on the decomposed layers.

Our results are reported in Table 2 where we compare to a wide variety of available compression benchmarks (results were adapted directly from the respective papers). The middle part and bottom part of the table for each network are organized into low-rank compression and filter pruning approaches, respectively. Note that the reported differences in accuracy ($\Delta$-Top1 and $\Delta$-Top5) are relative to our baseline accuracies. On ResNet18 we can reduce the number of FLOPs by 65% with minimal drop in accuracy compared to the best competing method (MUSCO, 58.67%). With a slightly higher drop in accuracy (-1.37%) we can even compress 76% of FLOPs. On AlexNet, our framework finds networks with -0.21% and -0.41% difference in accuracy with over 77% and 81% fewer FLOPs. This constitutes a more-than-10% improvement in terms of FLOPs compared to current state-of-the-art (L-Rank) for similar accuracy drops.

### 3.3 Ablation Study

To investigate the different features of our method we ran compression experiments using multiple variations derived from our method, see Figure 7. For the simplest version of our method we consider a constant per-layer compression ratio and fix the value of $k$ to either 3 or 5 for all layers denoted by ALDS-Simple3 and ALDS-Simple5, respectively. Note that ALDS-Simple with $k = 1$ corresponds to the SVD comparison method. For the version denoted by ALDS-Error3 we fix the number of subspaces per layer ($k = 3$) and only run the global step of ALDS (Line 4 of Algorithm 1) to determine the optimal per-layer compression ratio. The results of our ablation study in Figure 7 indicate that our method clearly benefits from the combination of both the global and local step in terms of the number of subspaces ($k$) and the rank per subspace ($j$).

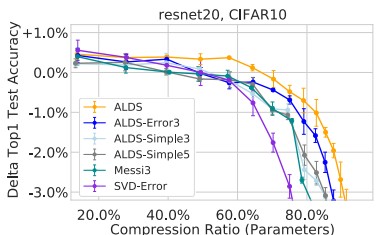

Figure 7: One-shot ablation study of ALDS for Resnet20 (CIFAR10).

We also compare our subspace clustering (channel slicing) to the clustering technique of Maalouf et al. (2021), which clusters the matrix columns using projective clustering. Specifically, we replace the channel slicing of ALDS-Simple3 with projective clustering (Messi3 in Figure 7). As expected Messi improves the performance over ALDS-Simple but only slightly and the difference is essentially negligible. Together with the computational disadvantages of Messi-like clustering methods (unstructured, NP-hard; see Section 2.1) ALDS-based simple channel slicing is therefore the preferred choice in our context.

## 4 Related Work

Our work builds upon prior work in neural network compression. We discuss related work focusing on pruning, low-rank compression, and global aspects of compression.

**Unstructured pruning.** Weight pruning (Lin et al., 2020b; Molchanov et al., 2016, 2019; Singh and Alistarh, 2020; Wang et al., 2021; Yu et al., 2018) techniques aim to reduce the number of individual weights, e.g., by removing weights with absolute values below a threshold (Han et al., 2015; Renda et al., 2020), or by using a mini-batch of data points to approximate the influence of each parameter on the loss function (Baykal et al., 2019a,b). However, since these approaches generate sparse instead of smaller models they require some form of sparse linear algebra support for runtime speed-ups.

**Structured pruning.** Pruning structures such as filters directly shrinks the network (Chen et al., 2020; Li et al., 2019b; Lin et al., 2020a; Liu et al., 2019a; Luo and Wu, 2020; Ye et al., 2018). Filters can be pruned using a score for each filter, e.g., weight-based (He et al., 2018, 2017) or data-informed (Liebenwein et al., 2020; Yu et al., 2018), and removing those with a score below a threshold. It is worth noting that filter pruning is complimentary to low-rank compression.

**Low-rank compression (local step).** A common approach to low-rank compression entails tensor decomposition including Tucker-decomposition (Kim et al., 2015b), CP-decomposition (Lebedev et al., 2015), Tensor-Train (Garipov et al., 2016; Novikov et al., 2015) and others (Denil et al., 2013; Ioannou et al., 2017; Jaderberg et al., 2014). Other decomposition-like approaches include weight

sharing, random projections, and feature hashing (Arora et al., 2018; Chen et al., 2015a,b; Shi et al., 2009; Ullrich et al., 2017; Weinberger et al., 2009). Alternatively, low-rank compression can be performed via matrix decomposition (e.g., SVD) on flattened tensors as done by Denton et al. (2014); Sainath et al. (2013); Tukan et al. (2020); Xue et al. (2013); Yu et al. (2017) among others. Chen et al. (2018); Denton et al. (2014); Maalouf et al. (2021) also explores the use of subspace clustering before applying low-rank compression to each cluster to improve the approximation error. Notably, most prior work relies on some form of expensive approximation algorithm – even to just solve the per-layer low-rank compression, e.g., clustering or tensor decomposition. In this paper, we instead focus on the global compression problem and show that simple compression techniques (SVD with channel slicing) are advantageous in this context as we can use them as efficient subroutines. We note that we can even extend our algorithm to multiple, different types of per-layer decomposition.

**Network-aware compression (global step).** To determine the rank (or the compression ratio) of each layer, prior work suggests to account for compression during training (Alvarez and Salzmann, 2017; Ioannou et al., 2016, 2015; Wen et al., 2017; Xu et al., 2020), e.g, by training the network with a penalty that encourages the weight matrices to be low-rank. Others suggest to select the ranks using variational Bayesian matrix factorization (Kim et al., 2015b). In their recent paper, Chin et al. (2020) suggest to produce an entire set of compressed networks with different accuracy/speed trade-offs. Our paper was also inspired by a recent line of work towards automatically choosing or learning the rank of each layer (Gusak et al., 2019; Idelbayev and Carreira-Perpinán, 2020; Li and Shi, 2018; Tiwari et al., 2021; Zhang et al., 2015b,c). We take such approaches further and suggest a global compression framework that incorporates multiple decomposition techniques with more than one hyper-parameter per layer (number of subspaces and ranks of each layer). This approach increases the number of local minima in theory and helps improving the performance in practice.

# 5 Discussion and Conclusion

**Practical benefits.** By conducting a wide variety of experiments across multiple data sets and networks we have shown the effectiveness and versatility of our compression framework compared to existing methods. The runtime of ALDS is negligible compared to retraining and it can thus be efficiently incorporated into compress-retrain pipelines.

**ALDS as modular compression framework.** By separately considering the low-rank compression scheme for each layer (local step) and the actual low-rank compression (global step) we have provided a framework that can efficiently search over a set of desired hyperparameters that describe the low-rank compression. Naturally, our framework can thus be generalized to other compression schemes (such as tensor decomposition) and we hope to explore these aspects in future work.

**Error bounds lead to global insights.** At the core of our contribution is our error analysis that enables us to link the global and local aspects of layer-wise compression techniques. We leverage our error bounds in practice to compress networks more effectively via an automated rank selection procedure without additional tedious hyperparameter tuning. However, we also have to rely on a proxy definition (maximum relative error) of the compression error to enable a tractable solution that we can implement efficiently. We hope these observations invigorate future research into compression techniques that come with tight error bounds – potentially even considering retraining – which can then naturally be wrapped into a global compression framework.

# Acknowledgments

This research was sponsored by the United States Air Force Research Laboratory and the United States Air Force Artificial Intelligence Accelerator and was accomplished under Cooperative Agreement Number FA8750-19-2-1000. The views and conclusions contained in this document are those of the authors and should not be interpreted as representing the official policies, either expressed or implied, of the United States Air Force or the U.S. Government. The U.S. Government is authorized to reproduce and distribute reprints for Government purposes notwithstanding any copyright notation herein. This work was further supported by the Office of Naval Research (ONR) Grant N00014-18-1-2830.

## Funding Transparency Statement

Authors declare no competing interests. *Funding in direct support of this work:* the United States Air Force Research Laboratory and the United States Air Force Artificial Intelligence Accelerator accomplished under Cooperative Agreement Number FA8750-19-2-1000 and the Office of Naval Research (ONR) Grant N00014-18-1-2830.

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
