# Supplementary Material:
# "Compressing Neural Networks: Towards Determining the Optimal Layer-wise Decomposition"

**Lucas Liebenwein**[*]
MIT CSAIL
lucas@csail.mit.edu

**Alaa Maalouf**[*]
University of Haifa
alaamalouf12@gmail.com

**Dan Feldman**
University of Haifa
dannyf.post@gmail.com

**Daniela Rus**
MIT CSAIL
rus@csail.mit.edu

## Contents

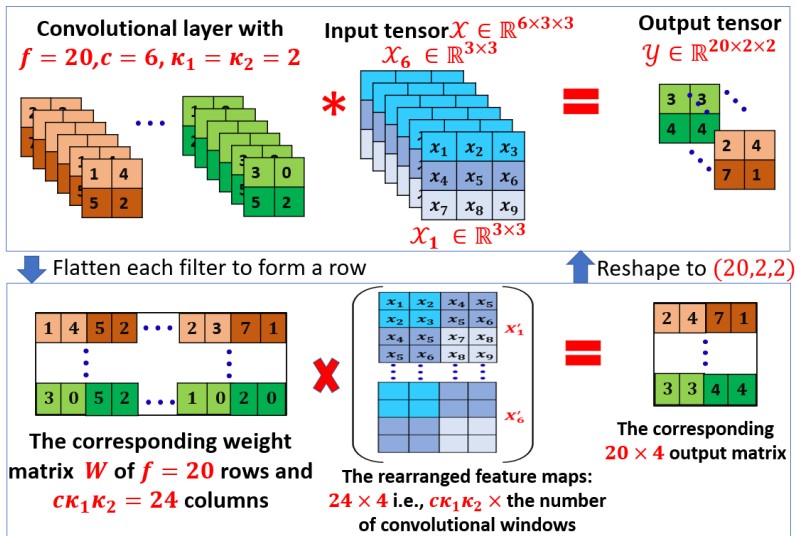

Figure 8: **Convolution to matrix multiplication.** A convolutional layer of $f = 20$ filters, $c = 6$ channels, and $2 \times 2$ kernel ($\kappa_1 = \kappa_2 = 2$). The input tensor shape is $6 \times 3 \times 3$. The corresponding weight matrix has $f = 20$ rows (one row per filter) and 24 columns ($c \times \kappa_1 \times \kappa_2$), as for the corresponding feature matrix, it has 24 rows and 4 columns, the 4 here is the number of convolution windows (i.e., number of pixels/entries in each of the output feature maps). After multiplying those matrices, we reshape them to the desired shape to obtain the desired output feature maps.

## A    Further Method Details

In this section, we provide more details pertaining to our method.

### A.1    Method Preliminaries

Our layer-wise compression technique hinges upon the insight that any linear layer may be cast as a matrix multiplication, which enables us to rely on SVD as compression subroutine. Focusing on convolutions we show how such a layer can be recast as matrix multiplication. Similar approaches have been used by Denton et al. (2014); Idelbayev and Carreira-Perpiñán (2020); Wen et al. (2017) among others.

**Convolution to matrix multiplication.**    For a given convolutional layer of $f$ filters, $c$ channels, $\kappa_1 \times \kappa_2$ kernel and an input feature map with $c$ features, each of size $m_1 \times m_2$, we denote by $\mathcal{W} \in \mathbb{R}^{f \times c \times \kappa_1 \times \kappa_2}$ and $\mathcal{X} \in \mathbb{R}^{c \times m_1 \times m_2}$ the weight tensor and input tensor, respectively. Moreover, let $W \in \mathbb{R}^{f \times c\kappa_1\kappa_2}$ denote the unfolded matrix operator of the layer constructed from $\mathcal{W}$ by flattening the $c$ kernels of each filter into a row and stacking the rows to form a matrix. Finally, let $p$ denote the total number of sliding blocks and $X \in \mathbb{R}^{c\kappa_1\kappa_2 \times p}$ denote the unfolded input matrix, which is constructed from the input tensor $\mathcal{X}$ as follows: while simulating the convolution by sliding $\mathcal{W}$ along $\mathcal{X}$ we extract the sliding local blocks of $\mathcal{X}$ across all channels by flattening each block into a $c\kappa_1\kappa_2$-dimensional column vector and concatenating them together to form $X$. As illustrated in Figure 8 we may now express the convolution $\mathcal{Y} = \mathcal{W} * \mathcal{X}$ as the matrix multiplication $Y = WX$, where $\mathcal{Y} \in \mathbb{R}^{f \times p_1 \times p_2}$ and $Y \in \mathbb{R}^{f \times p}$ correspond to the tensor and matrix representation of the output feature maps, respectively, and $p_1$, $p_2$ denote the spatial dimensions of $\mathcal{Y}$. The equivalence of $\mathcal{Y}$ and $Y$ can be easily established via an appropriate reshaping operation since $p = p_1 p_2$.

**Efficient tensor decomposition via SVD.**    Equipped with the notion of correspondence between convolution and matrix multiplication our goal is to decompose the layer via its matrix operator $W \in \mathbb{R}^{f \times c\kappa_1\kappa_2}$. To this end, we compute the $j$-rank approximation of $W$ using SVD and factor it into a pair of smaller matrices $U \in \mathbb{R}^{f \times j}$ and $V \in \mathbb{R}^{j \times c\kappa_1\kappa_2}$. More details on how to compute $U$ and $V$ are given in Section A. We may then replace the original convolution, represented by $W$, by two smaller convolutions, represented by $V$ and $U$ for the first and second layer, respectively. Just like for the original layer, we can establish an equivalent convolution layer for both $U$ and $V$ as depicted in Figure 9. To establish the equivalence we note that (a) every row of the matrices $V$ and

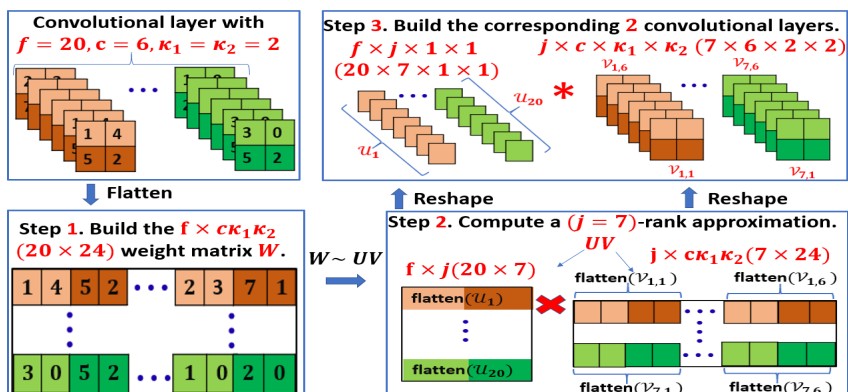

Figure 9: **Low-rank decomposition for convolutional layers via SVD.** The given convolution, c.f. Figure 8, has 20 filters, each of shape $6 \times 2 \times 2$, resulting in a total of $480$ parameters. After extracting the corresponding weight matrix $W \in \mathbb{R}^{20 \times 24}$ ($f \times c\kappa_1\kappa_2$), we compute its ($j = 7$)-rank decomposition to obtain the pair of matrices $U \in \mathbb{R}^{20 \times 7}$ ($f \times j$) and $V \in \mathbb{R}^{7 \times 24}$ ($j \times c\kappa_1\kappa_2$). Those matrices are encoded back as a pair of convolutional layers, the first (corresponding to $V$) has $j = 7$ filters, $c = 6$ channels and a $2 \times 2$ ($\kappa_1 \times \kappa_2$) kernel, whereas the second (corresponding to $U$) is a $1 \times 1$ convolution of $f = 20$ filters, and $j = 7$ channels. The resulting layers have $308$ parameters.

$U$ corresponds to a flattened filter of the respective convolution, and (b) the number of channels in each layer is equal to the number of channels in its corresponding input tensor. Hence, the first layer, which is represented by $V \in \mathbb{R}^{j \times c\kappa_1\kappa_2}$ has $j$ filters, c.f. (a), each consisting of $c$ channels, c.f. (b), with kernel size $\kappa_1 \times \kappa_2$. The second layer corresponding to $U$ has $f$ filters, c.f. (a), $j$ channels, c.f. (b), and a $1 \times 1$ kernel and may be equivalently represented as the tensor $\mathcal{U} \in \mathbb{R}^{f \times j \times 1 \times 1}$. Note that the number of weights is reduced from $fc\kappa_1\kappa_2$ to $j(f + c\kappa_1\kappa_2)$.

### A.2 Clustering Methods

As previously explained in Section 2.1, one can cluster the columns of the corresponding weight matrix $W$, instead of clustering the channels of the convolutional layer. Here, the channel clustering can be defined as constraint clustering of these columns, where columns which include entries that correspond to the same kernel (e.g., the first $4$ columns in $W$ from Figure 9) are guaranteed to be in the same cluster.

This generalization is easily adaptable to other clustering methods that generate a wider set of solutions, e.g., the known $k$-**means**. An intuitive choice for our case is **projective clustering** and its variants. The goal of projective clustering is to compute a set of $k$ subspaces, each of dimension $j$, that minimizes the sum of *squared* distances from each column in $W$ to its *closest* subspace from this set. Then, we can partition the columns of $W$ into $k$ subsets according to their nearest subspace from this set. This is a natural extension of SVD that solves this problem for the case of $k = 1$. However, this problem is known to be NP-hard, hence expensive approximation algorithms are required to solve it, or alternatively, a local minimum solution can be obtained using the Expectation-Maximization method (EM) (Dempster et al., 1977).

A more robust version of the previous method is to minimize the sum of *non-squared* distances from the points to the subspaces. This approach is known to be more robust toward outliers ("far away points"). Similarly to the original variant (the case of squared distance), we can use the EM method to obtain a "good guess" for a solution of this problem, however, the EM method requires an algorithm that solves the problem for the case of $k = 1$, i.e., computing the subspace that minimizes the sum of (non-squared) distances from those columns (for the sum of squared distances case, SVD is this algorithm). Unfortunately, there is only approximation algorithms (Clarkson and Woodruff, 2015; Tukan et al., 2020) for this case, and the deterministic versions are expensive in terms of running time.

Furthermore, and probably more importantly, all of these methods cannot be considered as structured compression since arbitrary clustering may require re-shuffling the input tensors which could lead to significant slow-downs during inference. For example, when compressing a fully-connected layer,

the arbitrary clustering may result in nonconsecutive neurons from the first layer that are connected to the same neuron in the second layer, while neurons that are between them are not. Hence, these layers can only have a large, sparse instead of a small, dense representation.

To this end, we choose to use **channel slicing**, i.e., we simply split the channels of the convolutional layer into $k$ chunks, where each chunk has at most $c/k$ consecutive channels. Splitting the channels into consecutive subsets (without allowing any arbitrary clustering) and applying the factorization on each one results in a structurally compressed layer without the need of special software/hardware support. Furthermore, this approach is the fastest among all the others. Finally, while other approaches may give a better initial guess for a compressed network in theory, in practice this is not the case; see Figures 10. This is due to the global compression framework (Section 2.2) which repeatedly utilizes the channel slicing.

We see that in practice, our method improve upon state-of-the-art techniques and obtains smaller networks with higher accuracy without the use of those complicated approaches that may result in sparse but not smaller network; see Section C.3.

### A.3 Compressing via SVD

As explained in Section 2.1, the weights of a convolutional layer (or a dense layer) can be encoded into a matrix $W \in \mathbb{R}^{f \times ck_1k_2}$, where $f$ and $c$ are the number of filters and channels in the layer, respectively, and $k_1 \times k_2$ is the size of the kernel. In order to compress the matrix $W$ (and thus its corresponding layer) we aim to factor it into a pair of smaller matrices $U \in \mathbb{R}^{f \times j}$ and $V \in \mathbb{R}^{j \times ck_1k_2}$, such that $UV$ approximates the original matrix operator $W$.

**How to compute the matrices $U$ and $V$?** For simplicity, let $d = ck_1k_2$. We factor the matrix $W \in \mathbb{R}^{f \times d}$ via SVD to obtain $W = \tilde{U}\tilde{\Sigma}\tilde{V}$, where $\tilde{U}$ is an $f \times f$ orthogonal matrix, $\tilde{\Sigma}$ is an $f \times d$ rectangular diagonal matrix with non-negative real numbers on the diagonal, and $\tilde{V}$ is an $d \times d$ orthogonal matrix.

To compute a $j$-rank approximation of $W$, we can simply define the following 3 matrices: $U \in \mathbb{R}^{f \times j}$, $\overline{V} \in \mathbb{R}^{j \times d}$, and $\Sigma \in \mathbb{R}^{j \times j}$, where $U$ is constructed by taking the first $j$ columns of $\tilde{U}$, $\overline{V}$ by taking the first $j$ rows of $\tilde{V}$, and $\Sigma$ is a diagonal matrix such that the $j$ entries on its diagonal are equal to the first $j$ entries on the diagonal of $\tilde{\Sigma}$. Now we have that $U\Sigma\overline{V}$ is the $j$-rank approximation of $W$.

Finally, we can define $V = \Sigma\overline{V}$ to obtain a factorization of the $j$-rank approximation of $W$ as $UV$.

### A.4 Efficient Implementation of ALDS (Algorithm 1)

Our algorithm that is suggested in Section 2.2 aims at minimizing the maximum relative error $\varepsilon := \max_{\ell \in [L]} \varepsilon^\ell$ across the $L$ layers of the network as a proxy for the true cost, where $\varepsilon^\ell$ is the theoretical maximum relative error in the $\ell^{\text{th}}$:

$$\varepsilon^\ell := \frac{\left\| \hat{W}^\ell - W^\ell \right\|}{\| W^\ell \|}.$$

Through Algorithm 1, for every $\ell \in [L]$ we need to repeatedly compute $\varepsilon^\ell$ as a function of $j^\ell$ and $k^\ell$. At Line 4, we are given a guess for the optimal values of $k^1, \ldots, k^L$, and our goal is to compute the values $j^1, \ldots, j^L$ such that the resulting errors $\varepsilon^1, \ldots, \varepsilon^L$ are (approximately) equal in order to minimize the maximum error $\max_{\ell \in [L]} \varepsilon^\ell$ while achieving the desired global compression ratio. To this end, we guess a value for $\varepsilon$ and for given $k^1, \ldots, k^L$ pick the corresponding $j^1, \ldots, j^L$ such that $\varepsilon$ constitutes a tight upper bound for the relative error in each layer. Based on the now resulting budget (and consequently compression ratio) we can now improve our guess of $\varepsilon$, e.g., via binary search or other types of root finding algorithms, until we convergence to a value of $\varepsilon$ that corresponds to our desired overall compression ratio.

Subsequently, for each layer we are given specific values of $k^\ell$ and $j^\ell$, which implies that we are given a budget $b^\ell$ for every layer $\ell \in [L]$. Subsequently, we re-assign the number of subspaces $k^\ell$ and their ranks $j^\ell$ for each layer by iterating through the finite set of possible values for $k^\ell$ (Line 7) and choosing the combination of $j^\ell$, $k^\ell$ that minimizes the relative error for the current layer budget $b^\ell$ (computed in Line 6).

We then iteratively repeat both steps until convergence (Lines 3-8).

Hence, instead of computing the cost of each layer at each step, we can save a lookup table that stores the errors $\varepsilon^\ell$ for the possible values of $k^\ell$ and $j^\ell$ of each layer. For every layer $\ell \in [L]$, we iterate over the finite set of values of $k^\ell$, and we split the matrix $W^\ell$ to $k^\ell$ matrices (according to the channel slicing approach that is explained in Section 2.1), then we compute the SVD factorization of each matrix from these $k^\ell$ matrices, and finally, compute $\varepsilon^\ell$ that corresponds to a specific $j^\ell$ ($k^\ell$ is already given) in $O(fd)$ time, where $f$ is the number of rows in the weight matrix that corresponds to the $\ell$th layer and $d$ is the number of columns.

Furthermore, instead of computing each option of $\varepsilon^\ell$ in $O(fd)$ time, we use the upper bound derived in Theorem 1 to compute it in $O(k)$ time and saving it in the lookup table. Specifically, we can express the relative error as a function of the rank and we thus only need to solve the underlying SVD for each layer once for each value of $k^\ell$. Without Theorem 1 we would need to compute the relative error (operator norm) for each pair $j^\ell, k^\ell$ separately. This would in turn result in a significant slowdown of the runtime of Algorithm 1. Hence, the combined use of a look-up table and the application of Theorem 1 ensures a more efficient implementation of Algorithm 1.

### A.5 Additional Discussion of ALDS (Algorithm 1)

Below, we include additional details and clarification regarding Algorithm 1.

**Overview**

At a high-level, Algorithm 1 aims to find a local optimum for the optimization procedure described in Equation (8). We hereby iteratively optimize for $k^1, \ldots, k^L$ and $j^1, \ldots, j^L$. The step where we fix the set of $k$'s and optimize for the set of $j$'s is Line 4, whereas in Line 7 we fix the layer budget and optimize for the set of $k$'s. At each step the objective is minimized. Thus for a fixed seed, ALDS converges to a local optimum of (8). We then repeat the entire procedure multiple times with different random seeds to improve the quality of the local optimum.

**Line 4: OPTIMALRANKS($\mathbf{CR}, k^1, \ldots, k^L$)**

At this step we are given a guess for the optimal values of $k^1, \ldots, k^L$, and our goal is to compute the values $j^1, \ldots, j^L$ that minimize the objective function described in Equation (8), i.e., the maximum error $\max_{\ell \in [L]} \varepsilon^\ell$, while achieving the desired global compression ratio $\mathbf{CR}$.

To find the optimal solution, we note the following. Recall that $k$'s are fixed.

1. **The maximum error is minimized exactly when all errors are equal.** To see that this is indeed the case we can proceed by contradiction. Suppose we found an optimal solution where all errors are not equal. Then we could use some of our compression budget to add more parameters to the layer with the maximum error while removing the same amount of parameters from the layer with minimum error. Since adding more parameters improves the error we just lowered the maximum error by adding more parameters to the layer with the maximum error. Hence, this leads to a contradiction proving our initial statement.

2. **A given constant error across layers corresponds to a fixed compression ratio.** This should be very straightforward to see. Specifically, for a given layer error we can find the corresponding rank and the rank implies how many parameters the compressed layer will have. This then implies a fixed compression ratio. Moreover, note that this relation is monotonic.

Both (1.) and (2.) together imply that we can use a binary search or some other root finding algorithm to determine the corresponding constant error for a desired compression ratio OPTIMALRANKS. The solution of our binary search will then be the corresponding set of $j$'s (ranks) for each layer that minimizes the maximum error for a desired compression ratio and given set of $k$'s (recall that we optimize for $k$'s separately).

**Line 7: OPTIMALSUBSPACES($b^\ell$)**

This step is fairly straightforward to follow. First, we note that for this step we proceed on a per-layer basis. Here, for each layer we are given specific values of $k^\ell$ and $j^\ell$, which implies that we are given a budget $b^\ell$ for every layer $\ell \in [L]$. Subsequently, we re-assign the number of subspaces $k^\ell$ and their

ranks $j^\ell$ for each layer $\ell$ as follow: We iterate through the finite set of possible values for $k^\ell$, for every such value $k^\ell$ we pick its corresponding $j^\ell$ such that the total size (number of parameters) of this layer is (approximately) the given budget $b^\ell$. Now, for every pair of candidates $k^\ell$ and $j^\ell$ we compute the relative error on this layer that is caused after compression with respect to these values. Finally, we choose the combination of $j^\ell$, $k^\ell$ that minimizes the relative error for the current layer budget $b^\ell$. We then discard the values found for $j^1, \ldots, j^L$ and re-optimize them in the next iteration of OPTIMALRANKS.

Note that for OPTIMALSUBSPACES there is no monotonic relation between the value of $k^\ell$ and the corresponding error like there is between the value of $j^\ell$ and the error. Hence, we proceed on a per-layer basis where we keep the per-layer budget constant during OPTIMALSUBSPACES as described above.

**Optimality**

From the details of the two steps, it should be very clear that the cost is decreasing at each step in the optimization procedure and we can thus conclude that for each random seed Algorithm 1 converges to a local optimum (at which point the cost will be non-increasing).

**Even More Remarks**

Note that above for OPTIMALRANKS we assumed that the errors ($\varepsilon^\ell$'s) are continuous but they are actually discrete given that they are a function of the rank which is discrete. However, as long as we can ensure that the objective decreases at every iteration we can still reach a local minimum.

Alternatively, we can solve the continuous relaxation of the above problem and use a randomized rounding[2] approach to get an approximately optimal solution.

In practice, however, we found that it is not necessary to add this additional complication step since it is sufficient that the cost objective decreases at every time step and we cannot hope to obtain a global optimum anyway (we can only approximate it with repeating the optimization procedure with multiple random seeds, which we do, see Algorithm 1).

### A.6 Extensions of ALDS

As mentioned in Section 2.3 ALDS can be readily extended to any desired *set* of low-rank compression techniques. Specifically, we can replace the local step of Line 7 by a search over different methods, e.g., Tucker decomposition, PCA, or other SVD compression schemes, and return the best method for a given budget. In general, we may combine ALDS with any low-rank compression as long as we can efficiently evaluate the per-layer error of the compression scheme. Note that this essentially equips us with a framework to automatically choose the per-layer decomposition technique fully automatically.

To this end, we test an extension of ALDS where in addition to searching over multiple values of $k^\ell$ we simultaneously search over various flattening schemes to convert a convolutional tensor to a matrix before applying SVD.

As before, let $\mathcal{W} \in \mathbb{R}^{f \times c \times \kappa_1 \times \kappa_2}$ denote the weight tensor for a convolutional layer with $f$ filters, $c$ input channels, and a $\kappa_1 \times \kappa_2$ kernel. Moreover, let $j$ denote the desired rank of the decomposition. We consider the following schemes to automatically search over:

- SCHEME 0: flatten the tensor to a matrix of shape $f \times c\kappa_1\kappa_2$. The decomposed layers correspond to a $j \times c \times \kappa_1 \times \kappa_2$-convolution followed by a $f \times j \times 1 \times 1$-convolution. This is the same scheme as used in ALDS.

- SCHEME 1: flatten the tensor to a matrix of shape $f\kappa_1 \times c\kappa_2$. The decomposed layers correspond to a $j \times c \times 1 \times \kappa_2$-convolution followed by a $f \times j \times \kappa_1 \times 1$-convolution.

- SCHEME 2: flatten the tensor to a matrix of shape $f\kappa_2 \times c\kappa_1$. The decomposed layers correspond to a $j \times c \times \kappa_1 \times 1$-convolution followed by a $f \times j \times 1 \times \kappa_2$-convolution.

- SCHEME 3: flatten the tensor to a matrix of shape $f\kappa_1\kappa_2 \times c$. The decomposed layers correspond to a $j \times c \times 1 \times 1$-convolution followed by a $f \times j \times \kappa_1 \times \kappa_2$-convolution.

---

[2]https://en.wikipedia.org/wiki/Randomized_rounding

We denote this method by ALDS+ and provide preliminary results in Section C.4. We note that since ALDS+ is a generalization of ALDS its performance is at least as good as the original ALDS. Moreover, our preliminary results actually suggest that the extension clearly improves upon the empirical performance of ALDS.

# B  Experimental Setup and Hyperparameters

Our experimental evaluations are based on a variety of network architectures, data sets, and compression pipelines. In the following, we provide all necessary hyperparameters to reproduce our experiments for each of the datasets and respective network architectures.

All networks were trained, compressed, and evaluated on a compute cluster with NVIDIA Titan RTX and NVIDIA RTX 2080Ti GPUs. The experiments were conducted with PyTorch 1.7 and our code is fully open-sourced [3].

All networks are trained according to the hyperparameters outlined in the respective original papers. During retraining, which is described in Section B.5, we reuse the same hyperparameters.

Moreover, each experiment is repeated 3 times and we report mean and mean, standard deviation in the tables and figures, respectively.

For each data set, we use the publicly available development set as test set and use a 90%/5%/5% split on the train set to obtain a separate train and *two* validation sets. One validation set is used for data-dependent compression methods, e.g., PCA (Zhang et al., 2015a); the other set is used for early stopping during training.

## B.1  Experimental Setup for CIFAR10

All relevant hyperparameters are outlined in Table 3. For each of the networks we use the training hyperparameters outlined in the respective original papers, i.e., as described by He et al. (2016), Simonyan and Zisserman (2014), Huang et al. (2017), and Zagoruyko and Komodakis (2016) for ResNets, VGGs, DenseNets, and WideResNets (WRN), respectively.

We add a warmup period in the beginning where we linearly scale up the learning rate from 0 to the nominal learning rate to ensure proper training performance in distributed training settings (Goyal et al., 2017).

During training we use the standard data augmentation strategy for CIFAR: (1) zero padding from 32x32 to 36x36; (2) random crop to 32x32; (3) random horizontal flip; (4) channel-wise normalization. During inference only the normalization (4) is applied.

The compression ratios are chosen according to a geometric sequence with the common ratio denoted by $\alpha$ in Table 3, i.e., the compression ratio for iteration $i$ is determined by $1 - \alpha^i$. The compression parameter $n_{\text{seed}}$ denotes the number of seeds used to initialize Algorithm 1 for compressing with PP.

## B.2  Experimental Setup for ImageNet

We report the relevant hyperparameters in Table 4. For ImageNet we consider the networks architectures Resnet18 (He et al., 2016), AlexNet (Krizhevsky et al., 2012), and MobileNetV2 (Sandler et al., 2018).

During training we use the following data augmentation: (1) randomly resize and crop to 224x224; (2) random horizontal flip; (3) channel-wise normalization. During inference, we use a center crop to 224x224 before (3) is applied.

Note that for MobileNetV2 we deploy a lower initial learning rate during retraining. Otherwise, all hyperparameters remain the same during retraining.

---

[3]Code repository: https://github.com/lucaslie/torchprune

Table 3: The experimental hyperparameters for training, compression, and retraining for the tested **CIFAR10** network architectures. "LR" and "LR decay" hereby denote the learning and the (multiplicative) learning rate decay, respectively, that is deployed at the epochs as specified. "$\{x, \ldots\}$" indicates that the learning rate is decayed every $x$ epochs.

| | | Hyperparameters | VGG16 | Resnet20 | DenseNet22 | WRN-16-8 |
|---|---|---|---|---|---|---|
| CIFAR10 | (Re-)Training | Test accuracy (%) | 92.81 | 91.4 | 89.90 | 95.19 |
| | | Loss | cross-entropy | cross-entropy | cross-entropy | cross-entropy |
| | | Optimizer | SGD | SGD | SGD | SGD |
| | | Epochs | 300 | 182 | 300 | 200 |
| | | Warm-up | 10 | 5 | 10 | 5 |
| | | Batch size | 256 | 128 | 64 | 128 |
| | | LR | 0.05 | 0.1 | 0.1 | 0.1 |
| | | LR decay | 0.5@$\{30, \ldots\}$ | 0.1@$\{91, 136\}$ | 0.1@$\{150, 225\}$ | 0.2@$\{60, \ldots\}$ |
| | | Momentum | 0.9 | 0.9 | 0.9 | 0.9 |
| | | Nesterov | ✗ | ✗ | ✓ | ✓ |
| | | Weight decay | 5.0e-4 | 1.0e-4 | 1.0e-4 | 5.0e-4 |
| | Compression | $\alpha$ | 0.80 | 0.80 | 0.80 | 0.80 |
| | | $n_{\text{seed}}$ | 15 | 15 | 15 | 15 |

Table 4: The experimental hyperparameters for training, compression, and retraining for the tested **ImageNet** network architectures. "LR" and "LR decay" hereby denote the learning and the (multiplicative) learning rate decay, respectively, that is deployed at the epochs as specified. "$\{x, \ldots\}$" indicates that the learning rate is decayed every $x$ epochs.

| | | Hyperparameters | ResNet18 | AlexNet | MobileNetV2 |
|---|---|---|---|---|---|
| ImageNet | (Re-)Training | Top 1 Test accuracy (%) | 69.64 | 57.30 | 71.85 |
| | | Top 5 Test accuracy (%) | 88.98 | 80.20 | 90.33 |
| | | Loss | cross-entropy | cross-entropy | cross-entropy |
| | | Optimizer | SGD | SGD | RMSprop |
| | | Epochs | 90 | 90 | 300 |
| | | Warm-up | 5 | 5 | 0 |
| | | Batch size | 256 | 256 | 768 |
| | | LR | 0.1 | 0.1 | 0.045 (1e-4) |
| | | LR decay | 0.1@$\{30, 60, 80\}$ | 0.1@$\{30, 60, 80\}$ | 0.98 per step |
| | | Momentum | 0.9 | 0.9 | 0.9 |
| | | Nesterov | ✗ | ✗ | ✗ |
| | | Weight decay | 1.0e-4 | 1.0e-4 | 4.0e-5 |
| | Compression | $\alpha$ | 0.80 | 0.80 | 0.80 |
| | | $n_{\text{seed}}$ | 15 | 15 | 15 |

## B.3 Experimental Setup for Pascal VOC

In addition to CIFAR and ImageNet, we also consider the segmentation task from Pascal VOC 2012 Everingham et al. (2015). We augment the nominal data training data using the extra labels as provided by Hariharan et al. (2011). As network architecture we consider a DeeplabV3 Chen et al. (2017) with ResNet50 backbone pre-trained on ImageNet.

During training we use the following data augmentation pipeline: (1) randomly resize (256x256 to 1024x1024) and crop to 513x513; (2) random horizontal flip; (3) channel-wise normalization. During inference, we resize to 513x513 exactly before the normalization (3) is applied.

We report both intersection-over-union (IoU) and Top1 test accuracy for each of the compressed and uncompressed networks. The experimental hyperparameters are summarized in Table 5.

## B.4 Baseline Methods

We implement and compare against the following compression methods for our baseline experiments:

1. PCA (Zhang et al., 2015a) decomposes each layer based on principle component analysis of the pre-activation (output of linear layer). We implement the symmetric, linear version of their method. The per-layer compression ratio is based on the greedy solution for minimizing the

Table 5: The experimental hyperparameters for training, compression, and retraining for the tested **VOC** network architecture. "LR" and "LR decay" hereby denote the learning and the learning rate decay, respectively. Note that the learning rate is polynomially decayed after each step.

| | Hyperparameters | | DeeplabV3-ResNet50 |
|---|---|---|---|
| | | IoU Test accuracy (%) | 69.84 |
| | | Top 1 Test accuracy (%) | 94.25 |
| | | Loss | cross-entropy |
| | | Optimizer | SGD |
| | (Re-)Training | Epochs | 45 |
| Pascal VOC 2012 – Segmentation | | Warm-up | 0 |
| | | Batch size | 32 |
| | | LR | 0.02 |
| | | LR decay | $(1 - \text{"step"}/\text{"total steps"})^{0.9}$ |
| | | Momentum | 0.9 |
| | | Nesterov | ✗ |
| | | Weight decay | 1.0e-4 |
| | Compression | $\alpha$ | 0.80 |
| | | $n_{\text{seed}}$ | 15 |

product of the per-layer energy, where the energy is defined as the sum of singular values in the compressed layer, see Equation (14) of Zhang et al. (2015a).

2. SVD-ENERGY (Alvarez and Salzmann, 2017; Wen et al., 2017) decomposes each layer via matrix folding akin to our SVD-based decomposition. The per-layer compression ratio is found by keeping the relative energy reduction constant across layers, where energy is defined as the sum of squared singular values.

3. SVD (Denton et al., 2014) decomposes each layer via matrix folding akin to our SVD-based decomposition. However, we hereby fix $k^\ell = 1$ for all layers $\ell \in [L]$ in order to provide a nominal comparison akin of "standard" tensor decomposition. The per-layer compression ratio is kept constant across all layers.

4. L-RANK (Idelbayev and Carreira-Perpinán, 2020) decomposes each layer via matrix folding akin to our SVD-based decomposition. The per-layer compression is determined by minimizing a joint cost objective of the energy and the computational cost of each layer, see Equation (5) of Idelbayev and Carreira-Perpinán (2020) for details.

5. FT Li et al. (2016) prunes the filters (or neurons) in each layer with the lowest element-wise $\ell_2$-norm. The per-layer compression ratio is set manually (constant in our implementation).

6. PFP Liebenwein et al. (2020) prunes the channels with the lowest sensitivity, where the data-dependent sensitivities are based on a provable notion of channel pruning. The per-layer prune ratio is determined based on the associated theoretical error guarantees.

## B.5 Compress-Retrain Pipeline

Recall that our baseline experiments are based on the following unified compress-retrain pipeline across all compression methods:

1. TRAIN for $e$ epochs according to the standard training schedule for the respective network.

2. COMPRESS the network according to the chosen method.

3. RETRAIN the network for the desired amount of $r$ epochs using the original training hyperparameters from the epochs in the range $[e - r, e]$.

4. ITERATIVELY repeat 1.-3. after projecting the decomposed layers back (**optional**).

In addition, we also consider experiments in the iterative learning rate rewinding setting, where steps 2 and 3 are repeated iteratively (optional step 4).

While various papers combine their compression methods with different retrain schedules we unify the compress-retrain pipeline across all tested methods for our baseline experiments to ensure that results are comparable. Note that the implemented compress-retrain pipeline as originally introduced by Renda et al. (2020) has been shown to yield consistently good compression results across various compression/pruning setups (unstructured, structured) and tasks (computer vision, NLP). Hence, we choose to concentrate on that particular pipeline.

## C   Additional Experimental Results

In this section, we provide additional results of our experimental evaluations.

### C.1   Complete Tables for One-shot Compression Experiments from Section 3.1

Table 6: The maximal compression ratio for which the drop in test accuracy is at most some pre-specified $\delta$ on CIFAR10. The table reports compression ratio in terms of parameters and FLOPs, denoted by CR-P and CR-F, respectively. When the desired $\delta$ was not achieved for any compression ratio in the range the fields are left blank. The top values achieved for CR-P and CR-F are bolded.

| | Model | Prune Method | $\delta = 0.0\%$ | | | $\delta = 0.5\%$ | | | $\delta = 1.0\%$ | | | $\delta = 2.0\%$ | | | $\delta = 3.0\%$ | | |
|---|---|---|---|---|---|---|---|---|---|---|---|---|---|---|---|---|---|
| | | | Top1 Acc. | CR-P | CR-F | Top1 Acc. | CR-P | CR-F | Top1 Acc. | CR-P | CR-F | Top1 Acc. | CR-P | CR-F | Top1 Acc. | CR-P | CR-F |
| CIFAR10 | ResNet20 Top1: 91.39 | ALDS | +0.09 | **64.58** | **55.95** | -0.47 | **74.91** | **67.86** | -0.68 | **79.01** | **71.59** | -1.88 | **87.68** | **83.23** | -2.59 | **89.65** | **85.32** |
| | | PCA | +0.16 | 39.98 | 38.64 | -0.11 | 49.88 | 48.67 | -0.58 | 58.04 | 57.21 | -1.41 | 70.54 | 70.78 | -2.11 | 75.23 | 76.01 |
| | | SVD-Energy | +0.14 | 40.22 | 39.38 | -0.21 | 49.88 | 49.08 | -0.83 | 57.95 | 57.15 | -1.52 | 64.76 | 64.10 | -2.17 | 70.47 | 70.01 |
| | | SVD | +0.24 | 14.36 | 15.34 | -0.29 | 39.81 | 38.95 | -0.90 | 49.19 | 50.21 | -1.08 | 57.47 | 57.80 | -2.88 | 70.14 | 71.31 |
| | | L-Rank | +0.14 | 15.00 | 29.08 | -0.44 | 28.71 | 54.89 | -0.44 | 28.71 | 54.89 | -1.56 | 49.87 | 72.57 | -2.82 | 64.81 | 80.80 |
| | | FT | +0.15 | 15.29 | 16.66 | -0.32 | 39.69 | 39.57 | -0.75 | 57.77 | 55.85 | -1.88 | 74.89 | 71.76 | -2.71 | 79.29 | 76.74 |
| | | PFP | +0.12 | 28.74 | 20.56 | -0.28 | 40.28 | 30.06 | -0.85 | 58.26 | 46.94 | -1.56 | 70.49 | 59.78 | -2.57 | 79.28 | 69.27 |
| | VGG16 Top1: 92.78 | ALDS | +0.29 | **94.89** | **83.94** | -0.11 | **95.77** | **86.23** | -0.52 | **97.01** | **88.95** | -0.52 | **97.03** | 88.95 | -0.52 | **97.03** | 88.95 |
| | | PCA | +0.47 | 87.74 | 81.05 | -0.02 | 89.72 | 85.84 | -0.02 | 89.72 | 85.84 | -1.12 | 91.37 | 89.57 | -1.12 | 91.37 | 89.57 |
| | | SVD-Energy | +0.35 | 79.21 | 78.70 | -0.08 | 82.57 | 81.32 | -0.83 | 87.74 | 85.36 | -1.22 | 89.71 | 87.13 | -2.08 | 91.37 | 88.58 |
| | | SVD | +0.29 | 70.35 | 70.13 | +0.29 | 70.35 | 70.13 | -0.74 | 75.18 | 75.13 | -1.58 | 82.58 | 82.39 | -1.58 | 82.58 | 82.39 |
| | | L-Rank | +0.35 | 82.56 | 69.67 | -0.35 | 85.38 | 75.86 | -0.35 | 85.38 | 75.86 | -0.35 | 85.38 | 75.86 | -0.35 | 85.38 | 75.86 |
| | | FT | +0.17 | 64.81 | 62.16 | -0.47 | 79.13 | 78.44 | -0.87 | 82.61 | 82.41 | -1.95 | 89.69 | **89.91** | -2.66 | 91.35 | **91.68** |
| | | PFP | +0.16 | 89.73 | 74.61 | -0.47 | 94.87 | 84.76 | -0.96 | 96.40 | 88.38 | -1.33 | 97.02 | **90.25** | -1.33 | 97.02 | 90.25 |
| | DenseNet22 Top1: 89.88 | ALDS | +0.17 | 48.85 | **51.90** | -0.32 | **56.84** | **61.98** | -0.54 | **63.83** | **69.68** | -1.87 | **69.67** | **74.48** | -1.87 | **69.67** | **74.48** |
| | | PCA | +0.20 | 14.67 | 34.55 | +0.20 | 14.67 | 34.55 | -0.73 | 28.83 | 57.02 | -0.73 | 28.83 | 57.02 | -2.75 | 40.51 | 70.03 |
| | | SVD-Energy | | | | -0.29 | 15.16 | 19.34 | -0.29 | 15.16 | 19.34 | -1.28 | 28.62 | 33.26 | -2.21 | 40.20 | 44.72 |
| | | SVD | +0.13 | 15.00 | 15.33 | +0.13 | 15.00 | 15.33 | -0.87 | 26.73 | 27.41 | -0.87 | 26.73 | 27.41 | -2.51 | 37.99 | 39.25 |
| | | L-Rank | +0.26 | 14.98 | 35.21 | +0.26 | 14.98 | 35.21 | -0.90 | 28.67 | 63.55 | -1.82 | 40.33 | 73.45 | -1.82 | 40.33 | 73.45 |
| | | FT | +0.15 | 15.49 | 16.70 | -0.24 | 28.33 | 29.50 | -0.24 | 28.33 | 29.50 | -1.46 | 51.10 | 51.03 | -2.40 | 64.12 | 63.09 |
| | | PFP | +0.00 | 28.68 | 32.60 | -0.44 | 40.24 | 43.37 | -0.70 | 49.67 | 51.94 | -1.36 | 58.20 | 58.21 | -2.43 | 65.17 | 64.50 |
| | WRN16-8 Top1: 95.21 | ALDS | +0.05 | 28.67 | 13.00 | -0.42 | **87.77** | 79.90 | -0.88 | **92.75** | 87.39 | -1.53 | **95.69** | **92.50** | -2.23 | **97.01** | **95.51** |
| | | PCA | +0.14 | 15.00 | 7.98 | -0.49 | 85.33 | **83.45** | -0.96 | 91.33 | **90.23** | -1.76 | 93.90 | **93.15** | -2.45 | 94.87 | 94.30 |
| | | SVD-Energy | +0.29 | 15.01 | 6.92 | -0.41 | 64.75 | 60.94 | -0.81 | 85.38 | 83.52 | -1.90 | 91.38 | 90.04 | -2.46 | 92.77 | 91.58 |
| | | SVD | | | | | | | -0.96 | 40.20 | 39.97 | -1.63 | 70.48 | 70.49 | -1.63 | 70.48 | 70.49 |
| | | L-Rank | +0.25 | 14.99 | 6.79 | -0.45 | 49.86 | 58.00 | -0.88 | 75.20 | 82.26 | -1.70 | 87.73 | 92.03 | -2.18 | 89.72 | 93.51 |
| | | FT | +0.03 | **64.54** | **61.53** | -0.32 | 82.33 | 75.97 | -0.95 | 89.70 | 83.52 | -1.78 | 94.91 | 90.82 | -2.86 | 96.42 | 93.33 |
| | | PFP | +0.05 | 57.92 | 54.74 | -0.44 | 85.33 | 80.68 | -0.77 | 89.71 | 85.16 | -1.69 | 95.65 | 92.60 | -2.40 | 96.96 | 94.36 |

Table 7: The maximal compression ratio for which the drop in test accuracy is at most $\delta = 1.0\%$ for ResNet20 (CIFAR10) for various amounts of retraining (as indicated). The table reports compression ratio in terms of parameters and FLOPs, denoted by CR-P and CR-F, respectively. When the desired $\delta$ was not achieved for any compression ratio in the range the fields are left blank. The top values achieved for CR-P and CR-F are bolded.

| | Model | Prune Method | $r = 0\% \, e$ | | | $r = 5\% \, e$ | | | $r = 10\% \, e$ | | | $r = 25\% \, e$ | | | $r = 50\% \, e$ | | | $r = 100\% \, e$ | | |
|---|---|---|---|---|---|---|---|---|---|---|---|---|---|---|---|---|---|---|---|---|---|
| | | | Top1 Acc. | CR-P | CR-F | Top1 Acc. | CR-P | CR-F | Top1 Acc. | CR-P | CR-F | Top1 Acc. | CR-P | CR-F | Top1 Acc. | CR-P | CR-F | Top1 Acc. | CR-P | CR-F |
| CIFAR10 | ResNet20 Top1: 91.39 | ALDS | -0.13 | **14.82** | **7.03** | -0.53 | **35.87** | **26.27** | -0.73 | **43.12** | **33.65** | -0.65 | **43.14** | **33.33** | -0.86 | **62.39** | **54.40** | -0.88 | **81.29** | **74.23** |
| | | PCA | | | | | | | -0.74 | 19.31 | 18.64 | -0.70 | 19.34 | 18.44 | -0.59 | 36.21 | 35.19 | -0.74 | 60.29 | 59.81 |
| | | SVD-Energy | | | | -0.64 | 14.99 | 14.09 | -0.70 | 19.61 | 18.81 | -0.59 | 19.61 | 18.81 | -0.73 | 43.46 | 42.49 | -0.46 | 55.25 | 54.59 |
| | | SVD | | | | | | | -0.83 | 14.36 | 15.34 | -0.58 | 14.36 | 15.34 | -0.69 | 28.21 | 29.11 | -0.77 | 51.58 | 51.52 |
| | | L-Rank | | | | | | | | | | -0.64 | 15.00 | 29.08 | -0.33 | 15.00 | 29.08 | -0.44 | 28.71 | 54.89 |
| | | FT | | | | | | | | | | -0.67 | 15.29 | 16.66 | -0.69 | 27.76 | 28.40 | -0.75 | 57.77 | 55.85 |
| | | PFP | | | | | | | | | | -0.77 | 14.88 | 9.61 | -0.83 | 32.71 | 23.85 | -0.54 | 52.89 | 42.04 |

Table 8: The maximal compression ratio for which the drop in test accuracy is at most some pre-specified $\delta$ on ResNet18 and MobileNetV2 (both ImageNet). The table reports compression ratio in terms of parameters and FLOPs, denoted by CR-P and CR-F, respectively. When the desired $\delta$ was not achieved for any compression ratio in the range the fields are left blank. The top values achieved for CR-P and CR-F are bolded.

| | Model | Prune Method | $\delta = 0.0\%$ | | | $\delta = 0.5\%$ | | | $\delta = 1.0\%$ | | | $\delta = 2.0\%$ | | | $\delta = 3.0\%$ | | |
|---|---|---|---|---|---|---|---|---|---|---|---|---|---|---|---|---|---|
| | | | Top1/5 Acc. | CR-P | CR-F | Top1/5 Acc. | CR-P | CR-F | Top1/5 Acc. | CR-P | CR-F | Top1/5 Acc. | CR-P | CR-F | Top1/5 Acc. | CR-P | CR-F |
| ImageNet | ResNet18 Top1: 69.62 Top5: 89.08 | ALDS | +0.08/+0.18 | **59.43** | **33.08** | -0.15/-0.03 | **66.73** | **44.14** | -0.97/-0.49 | **72.73** | **52.81** | -1.63/-0.76 | **77.62** | **60.46** | -2.53/-1.44 | **81.75** | **67.62** |
| | | PCA | | | | | | | -0.88/-0.43 | 9.97 | 12.02 | -1.84/-0.94 | 50.43 | 51.07 | -2.34/-1.23 | 59.38 | 60.08 |
| | | SVD-Energy | | | | | | | | | | -1.91/-0.93 | 50.47 | 51.46 | -2.82/-1.53 | 59.42 | 60.24 |
| | | SVD | | | | | | | | | | -1.53/-0.83 | 50.44 | 50.38 | -2.06/-1.03 | 59.36 | 59.33 |
| | | L-Rank | | | | | | | -0.72/-0.26 | 10.01 | 32.40 | -0.72/-0.26 | 10.01 | 32.40 | -2.30/-1.26 | 26.25 | 58.59 |
| | | FT | +0.17/+0.21 | 9.96 | 10.78 | +0.17/+0.21 | 9.96 | 10.78 | -0.66/-0.32 | 26.12 | 26.62 | -1.72/-0.81 | 39.58 | 37.89 | -2.82/-1.55 | 50.62 | 45.74 |
| | | PFP | +0.25/+0.33 | 10.04 | 7.72 | -0.38/-0.15 | 26.35 | 19.14 | -0.38/-0.15 | 26.35 | 19.14 | -0.38/-0.15 | 26.35 | 19.14 | -2.80/-1.84 | 50.41 | 37.59 |
| | MobileNetV2 Top1: 71.85 Top5: 90.33 | ALDS | | | | | | | | | | -1.53/-0.73 | **32.97** | 11.01 | -1.53/-0.73 | **32.97** | 11.01 |
| | | PCA | | | | | | | -0.87/-0.55 | **20.91** | 0.26 | -0.87/-0.55 | 20.91 | 0.26 | -0.87/-0.55 | 20.91 | 0.26 |
| | | SVD-Energy | | | | | | | | | | -1.27/-0.57 | 20.02 | 8.57 | -2.50/-1.45 | 32.72 | 20.83 |
| | | SVD | | | | | | | | | | | | | | | |
| | | L-Rank | | | | | | | | | | | | | | | |
| | | FT | | | | | | | | | | -1.73/-0.85 | 21.31 | **20.23** | -2.68/-1.46 | 32.75 | **28.23** |
| | | PFP | | | | | | | -0.97/-0.40 | 20.02 | **7.96** | -1.44/-0.51 | **32.74** | 13.49 | -2.17/-0.85 | **43.32** | 19.21 |

Table 9: The maximal compression ratio for which the drop in test accuracy is at most $\delta = 1.0\%$ for ResNet18 (ImageNet) for various amounts of retraining (as indicated). The table reports compression ratio in terms of parameters and FLOPs, denoted by CR-P and CR-F, respectively. When the desired $\delta$ was not achieved for any compression ratio in the range the fields are left blank. The top values achieved for CR-P and CR-F are bolded.

| | Model | Prune Method | $r = 0\% e$ | | | $r = 5\% e$ | | | $r = 10\% e$ | | | $r = 25\% e$ | | | $r = 50\% e$ | | | $r = 100\% e$ | | |
|---|---|---|---|---|---|---|---|---|---|---|---|---|---|---|---|---|---|---|---|---|---|
| | | | Top1/5 Acc. | CR-P | CR-F | Top1/5 Acc. | CR-P | CR-F | Top1/5 Acc. | CR-P | CR-F | Top1/5 Acc. | CR-P | CR-F | Top1/5 Acc. | CR-P | CR-F | Top1/5 Acc. | CR-P | CR-F |
| ImageNet | ResNet18 Top1: 69.62 Top5: 89.08 | ALDS | -0.54/-0.24 | **39.57** | **15.20** | -0.48/-0.24 | **50.46** | **23.70** | -0.72/-0.30 | **53.50** | **26.90** | -0.64/-0.31 | **61.90** | **36.58** | -0.40/-0.23 | **68.78** | **47.23** | -0.73/-0.31 | **70.73** | **49.85** |
| | | PCA | | | | | | | -inf/-inf | 3.33 | 3.99 | -0.80/-0.38 | 26.21 | 27.53 | -0.76/-0.51 | 39.53 | 40.45 | -inf/-inf | 6.65 | 8.14 |
| | | SVD-Energy | | | | -0.28/-0.14 | 10.00 | 11.05 | -0.25/-0.12 | 10.00 | 11.05 | -0.55/-0.25 | 26.24 | 27.14 | -0.66/-0.33 | 39.56 | 40.48 | -inf/-inf | 3.33 | 3.66 |
| | | SVD | | | | -0.32/-0.13 | 9.98 | 9.94 | -0.19/-0.07 | 9.98 | 9.94 | -0.71/-0.34 | 30.63 | 30.82 | -0.59/-0.32 | 39.53 | 39.51 | -inf/-inf | 3.33 | 3.31 |
| | | L-Rank | | | | | | | -inf/-inf | 3.34 | 10.88 | -0.40/-0.23 | 10.01 | 32.40 | -0.16/+0.03 | 10.01 | 32.40 | -0.72/-0.26 | 10.01 | 32.40 |
| | | FT | | | | | | | -inf/-inf | 3.36 | 3.75 | -0.21/-0.15 | 9.95 | 10.78 | -0.83/-0.46 | 26.29 | 26.57 | -0.66/-0.32 | 26.12 | 26.62 |
| | | PFP | | | | | | | -inf/-inf | 3.34 | 2.37 | -0.14/-0.13 | 9.96 | 7.72 | -0.37/-0.31 | 20.76 | 15.14 | -0.38/-0.15 | 26.35 | 19.14 |

Table 10: The maximal compression ratio for which the drop in test accuracy is at most some pre-specified $\delta$ on DeeplabV3-ResNet50 (Pascal VOC2012). The table reports compression ratio in terms of parameters and FLOPs, denoted by CR-P and CR-F, respectively. When the desired $\delta$ was not achieved for any compression ratio in the range the fields are left blank. The top values achieved for CR-P and CR-F are bolded.

| | Model | Prune Method | $\delta = 0.0\%$ | | | $\delta = 0.5\%$ | | | $\delta = 1.0\%$ | | | $\delta = 2.0\%$ | | | $\delta = 3.0\%$ | | |
|---|---|---|---|---|---|---|---|---|---|---|---|---|---|---|---|---|---|
| | | | IoU/Top1 Acc. | CR-P | CR-F | IoU/Top1 Acc. | CR-P | CR-F | IoU/Top1 Acc. | CR-P | CR-F | IoU/Top1 Acc. | CR-P | CR-F | IoU/Top1 Acc. | CR-P | CR-F |
| VOCSegmentation2012 | DeeplabV3-ResNet50 IoU: 68.16 Top1: 94.25 | ALDS | +0.14/-0.15 | **64.38** | **64.11** | +0.14/-0.15 | **64.38** | **64.11** | +0.14/-0.15 | **64.38** | **64.11** | -1.22/-0.36 | **71.36** | **70.89** | -2.76/-0.61 | **76.96** | **76.37** |
| | | PCA | | | | -0.26/-0.02 | 31.59 | 31.63 | -0.88/-0.24 | 55.68 | 55.82 | -1.74/-0.39 | 64.33 | 64.54 | -2.54/-0.46 | 71.29 | 71.63 |
| | | SVD-Energy | | | | | | | | | | -1.88/-0.47 | 31.61 | 32.27 | -2.78/-0.62 | 44.99 | 45.60 |
| | | SVD | +0.01/-0.02 | 14.99 | 14.85 | -0.28/-0.18 | 31.64 | 31.51 | -0.89/-0.25 | 45.02 | 44.95 | -1.97/-0.50 | 64.42 | 64.42 | -1.97/-0.50 | 64.42 | 64.42 |
| | | L-Rank | | | | -0.42/-0.09 | 44.99 | 45.02 | -0.42/-0.09 | 44.99 | 45.02 | -1.29/-0.33 | 55.74 | 56.01 | -2.50/-0.57 | 64.39 | 64.82 |
| | | FT | | | | | | | | | | | | | | | |
| | | PFP | +0.01/-0.05 | 31.79 | 30.62 | -0.49/-0.21 | 45.17 | 43.93 | -0.84/-0.32 | 55.78 | 54.61 | -0.84/-0.32 | 55.78 | 54.61 | -2.43/-0.61 | 64.47 | 63.41 |

## C.2 Complete ImageNet Benchmark Results from Section 3.2

Results are provided in Table 11.

## C.3 Ablation Study

In order to gain a better understanding of the various aspects of our method we consider an ablation study where we selectively turn off various features of ALDS. Specifically, we compare the full version of ALDS to the following variants:

1. ALDS-ERROR solves for the optimal ranks (Line 4 of Algorithm 1) for a desired set of values for $k^1, \ldots, k^L$. We test $k^\ell = 3$, $\forall \ell \in [L]$. This variant tests the benefits of varying the number of subspaces compared to fixing them to a desired value.

2. SVD-ERROR corresponds to ALDS-Error with $k^\ell = 1$, $\forall \ell \in [L]$. This variants tests the benefits of having multiple subspaces in the first places in the context error-based allocation of the per-layer compression ratio.

Table 11: AlexNet and ResNet18 Benchmarks on ImageNet. We report Top-1, Top-5 accuracy and percentage reduction in terms of parameters and FLOPs denoted by CR-P and CR-F, respectively. Best results with less than 0.5% accuracy drop are bolded.

| | Method | $\Delta$-Top1 | $\Delta$-Top5 | CR-P (%) | CR-F (%) |
|---|---|---|---|---|---|
| **ResNet18**, Top1, 5: 69.64%, 88.98% | ALDS (Ours) | +0.41 | +0.37 | 66.70 | 42.70 |
| | ALDS (Ours) | **-0.38** | **+0.04** | **75.00** | **64.50** |
| | ALDS (Ours) | -0.90 | -0.25 | 78.50 | 71.50 |
| | ALDS (Ours) | -1.37 | -0.56 | 80.60 | 76.30 |
| | MUSCO (Gusak et al., 2019) | **-0.37** | **-0.20** | N/A | 58.67 |
| | TRP1 (Xu et al., 2020) | -4.18 | -2.5 | N/A | 44.70 |
| | TRP1+Nu (Xu et al., 2020) | -4.25 | -2.61 | N/A | 55.15 |
| | TRP2+Nu (Xu et al., 2020) | -4.3 | -2.37 | N/A | 68.55 |
| | PCA (Zhang et al., 2015b) | -6.54 | -4.54 | N/A | 29.07 |
| | Expand (Jaderberg et al., 2014) | -6.84 | -5.26 | N/A | 50.00 |
| | PFP (Liebenwein et al., 2020) | -2.26 | -1.07 | 43.80 | 29.30 |
| | SoftNet (He et al., 2018) | -2.54 | -1.2 | N/A | 41.80 |
| | Median (He et al., 2019) | -1.23 | -0.5 | N/A | 41.80 |
| | Slimming (Liu et al., 2017) | -1.77 | -1.19 | N/A | 28.05 |
| | Low-cost (Dong et al., 2017) | -3.55 | -2.2 | N/A | 34.64 |
| | Gating (Hua et al., 2018) | -1.52 | -0.93 | N/A | 37.88 |
| | FT (He et al., 2017) | -3.08 | -1.75 | N/A | 41.86 |
| | DCP (Zhuang et al., 2018) | -2.19 | -1.28 | N/A | 47.08 |
| | FBS (Gao et al., 2018) | -2.44 | -1.36 | N/A | 49.49 |
| **AlexNet**, Top1, 5: 57.30%, 80.20% | ALDS (Ours) | +0.10 | +0.45 | 92.00 | 76.10 |
| | ALDS (Ours) | -0.21 | -0.36 | 93.0 | 77.9 |
| | ALDS (Ours) | **-0.41** | **-0.54** | **93.50** | **81.4** |
| | Tucker (Kim et al., 2015a) | N/A | -1.87 | N/A | 62.40 |
| | Regularize (Tai et al., 2015) | N/A | -0.54 | N/A | 74.35 |
| | Coordinate (Wen et al., 2017) | N/A | -0.34 | N/A | 62.82 |
| | Efficient (Kim et al., 2019) | -0.7 | -0.3 | N/A | 62.40 |
| | L-Rank (Idelbayev et al., 2020) | **-0.13** | **-0.13** | N/A | **66.77** |
| | NISP (Yu et al., 2018) | -1.43 | N/A | N/A | 67.94 |
| | OICSR (Li et al., 2019a) | -0.47 | N/A | N/A | 53.70 |
| | Oracle (Ding et al., 2019) | -1.13 | -0.67 | N/A | 31.97 |

3. ALDS-SIMPLE picks the ranks in each layer for a desired set of values of $k^1, \ldots, k^L$ such that the per-layer compression ratio is constant. We test $k^\ell = 3$, $\forall \ell \in [L]$, and $k^\ell = 5$, $\forall \ell \in [L]$. This variant tests the benefits of allocating the per-layer compression ratio according to the layer error compared to a simple constant heuristic.

4. MESSI proceeds like ALDS-Simple but replaces the subspace clustering with projective clustering (Maalouf et al., 2021). We test $k^\ell = 3$, $\forall \ell \in [L]$. This variant tests the disadvantages of having a simple subspace clustering technique (channel slicing) compared to using a more sophisticated technique.

We note that ALDS-Simple with $k^\ell = 1$, $\forall \ell \in [L]$ corresponds to the SVD comparison method from the previous sections.

We study the variations on a ResNet20 trained on CIFAR10 in two settings: compression only and one-shot compress+retrain. The results are presented in Figures 10. We highlight that the complete variant of our algorithm (ALDS) consistently outperforms the weaker variants providing empirical evidence on the effectiveness of each of the core components of ALDS.

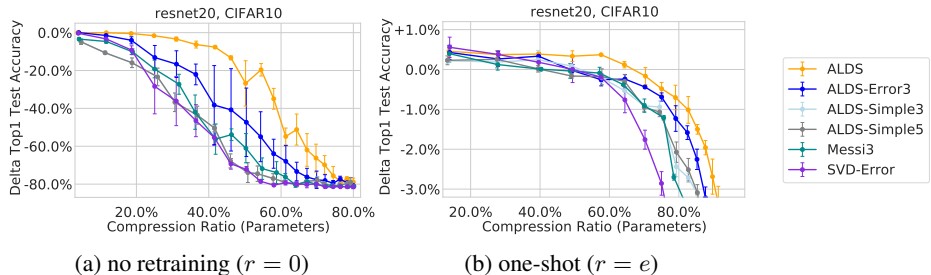

Figure 10: The difference in test accuracy ("Delta Top1 Test Accuracy") for various target compression ratios, ALDS-based/ALDS-related methods, and networks on CIFAR10.

We note that varying the number of subspaces for each layer in order to optimally assign a value of $k^\ell$ in each layer is crucial in improving our performance. This is apparent from the comparison between ALDS, ALDS-Error, and SVD-Error: having a fixed value for k yields sub-optimal results.

Picking an appropriate notion of cost (maximum relative error) is furthermore preferred over simple heuristics such a constant per-layer compression ratio. Specifically, the main difference between ALDS-Error and ALDS-Simple is the way how the ranks are determined for a given set of $k$'s: ALDS-Error optimizes for the error-based cost function while ALDS-Simple relies on a simple constant per-layer compression ratio heuristic. In practice, ALDS-Error outperforms ALDS-Simple across all tested scenarios.

Finally, we test the disadvantages of using a simple subspace clustering method. To this end, we compare ALDS-Simple and Messi for fixed values of $k$. While in some scenarios, particularly without retraining, Messi provides modest improvements over ALDS-Simple, the improvement is negligible for most settings. Moreover, note that Messi requires an expensive approximation algorithm as explained in Section A.2. This would in turn prevent us from incorporating Messi into the full ALDS framework in a computationally efficient manner. However, as apparent from the ablation study we exhibit the most performance gains for features related to global considerations instead of local, per-layer improvements. In addition, we should also note that Messi does not emit a structured reparameterization thus requires specialized software or hardware to obtain speed-ups. Consequently, we may conclude that channel slicing is the appropriate clustering technique in our context.

## C.4 Extensions of ALDS

We test and compare ALDS with ALDS+ (see Section A.6) to investigate the performance gains we can obtain from generalizing our local step to search over multiple decomposition schemes. We run one-shot compress-only experiments on ResNet20 (CIFAR10) and ResNet18 (ImageNet).

The results are shown in Figure 11. We find that ALDS+ can significantly increase the performance-size trade-off compared to our standards ALDS method. This is expected since by generalizing the local step of ALDS we are increasing the search space of possible decomposition solution. Using our ALDS framework we can efficiently and automatically search over the increased solution space. We envision that our observations will invigorate future research into the possibility of not only choosing the optimal per-layer compression ratio but also the optimal compression scheme.

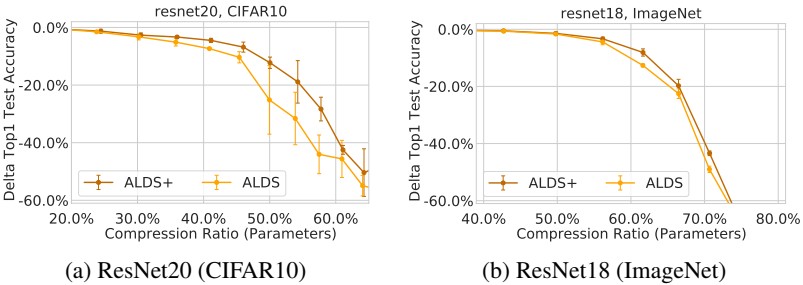

(a) ResNet20 (CIFAR10)  (b) ResNet18 (ImageNet)

Figure 11: The difference in test accuracy ("Delta Top1 Test Accuracy") for various target compression ratios, ALDS-based/ALDS-related methods, and networks on CIFAR10. The networks were compressed once and not retrained afterwards.