# OpenReview forum: "Compressing Neural Networks: Towards Determining the Optimal Layer-wise Decomposition"
_NeurIPS.cc/2021/Conference — NeurIPS 2021 Poster_

### Official Review · Reviewer_AgkJ · 2021-07-11

**Rating:** 5
**Confidence:** 3

**Summary:**

The paper propose a novel tensor decomposition pipeline. The propose automatic layer-wise decomposition (ALDC) method comprises two parts: local layer composition and global compression ratio optimization. For local layer decomposition, the weight tensor is first divided into groups and the optimization is within the group. For global compression ratio optimization, the author propose to calculate the relative error bound for each layer and then allocate a layer-wise computation budgets. The experiments are conducted on a various of datasets on image classification and segmentation. The proposed ALDC algorithm is verified over a wide range of network backbones.

**Limitations And Societal Impact:**

It is better to include some broader impacts in the paper.

**Main Review:**

Strong aspects:
1. The paper is well organized and the method is easy to follow.

2. The proposed method is well justified theoretically.

3. The experiments are comprehensive on the selected backbones and datasets. The results are promising.

4. More importantly, the proposed method do not need training of the network. This might be the main advantages over the NAS algorithms.

Weakness:
1. The selected backbones are all large and redundant. Experiments on efficient/mobile backbones (e.g. MBV2/EfficientNet/MobileNeXt) will make the proposed algorithm more convincing.

2. The main metric of the proposed method is based on the number of parameters. As decomposing a weight tensor will result a deeper network, I am concerned with the run time latency of the proposed method.

3. The reference format used by the authors is not correct.

I will increase my score if the author could address my concerns sufficiently.



**Time Spent Reviewing:**

8

---

> ### Author Response · Authors · 2021-08-10
> **Response to Reviewer AgkJ**
>
> ## Thank you
>
> We thank the reviewer for their time and effort in reviewing our paper and greatly appreciate the issues raised. We also appreciate the reviewer’s willingness to change their score if their concerns are sufficiently addressed.
>
> Below we provide a detailed response to all of the issues raised and we would be more than happy to further engage with the reviewer at any time during the discussion period to clear up remaining issues. Overall, we hope our response will encourage you to increase your score.
>
>
> ## 1. Experiments on efficient/mobile backbones
>
> #### **Current set of experiments**
>
> Thank you for this suggestion. We would like to note that our networks/backbones were chosen in such a way as to maximize the available results in literature to ensure the broadest comparison possible.
>
> #### **New experiments on MobileNetV2**
>
> However, since the rebuttal period we have started running additional compression experiments on MobileNetV2 (MBV2) for ImageNet as one of the efficient backbones suggested by the reviewer. In a separate comment, we are providing initial results while we are still testing additional compression ratios and hyperparameter configurations on this architecture. As we obtain more results we will continue to update our response.
>
> #### **Comparison methods and rolling updates**
>
> So far we could not identify any related low-rank compression work that conducted experiments on MBV2 for ImageNet and thus cannot provide comparisons. In order to still be able to compare the performance of our method, we plan on running some of our baseline comparisons before the end of the discussion period. Currently, we cannot run these experiments as we do not have the necessary GPUs available to run both our compression algorithm and others simultaneously. We will keep you posted.
>
>
> ## 2. Metrics beyond number of parameters
>
> We thank the reviewer for raising this issue and we fully agree that it is important to consider metrics beyond parameter count to assess the efficiency of the proposed compression method.
>
> #### **We report number of parameters and number of FLOPs**
>
> To this end, we would like to note that we already include both compression ratio in terms of number of parameters (`CR-P` in the paper) as well compression ratio in terms of floating point operations (FLOPs) (`CR-F` in the paper). Specifically, Table 2 reports `CR-F`. Moreover, in Section C of the appendix, all results are provided in the form of tables that report both `CR-P` and `CR-F`.
>
> #### **We can provide additional figures if desired**
>
> Currently, our figures are with respect to the reduction in parameters (`CR-P`). However, we can also generate the same figures with the x-axis being `CR-F`.
>
> Please let us know during the up-coming discussion period if you would like to see these figures. Then, we can provide an anonymous link to review the material and we can also incorporate them into our revised manuscript once we are allowed to update the paper.
>
> #### **Runtime**
>
> Finally, in terms of runtime latency, we would like to note that our compression method emits a dense reparameterization instead of some type of unstructured sparsity. Moreover, our compression scheme (including channel slicing) is consistent with the “grouped convolution” implementation in PyTorch and other deep-learning frameworks (see lines 90-92). Hence, we can expect direct speed-ups without any specialized software support.
>
> ## 3. Reference Format
>
> #### **We follow the official formatting instructions**
>
> We thank the reviewer for checking our paper with regards to the style guide as well. However, we are not quite sure what is wrong about the reference format. As far as we know, NeurIPS allows both author/year or numeric citation style (see Section 4 of [Formatting Instructions For NeurIPS 2021](https://media.neurips.cc/Conferences/NeurIPS2021/Styles/neurips_2021.pdf)).
>
> #### **Please informs us of any specific inconsistencies**
> Is there something else that is inconsistent or wrong with our reference/citation format? If so, please do let us know so that we can correct it once we are allowed to update the paper. Thank you!
>
> ## Final remarks
>
> We would like to thank the reviewer again for the valuable feedback. We are happy to be able to address all their comments and hope that our response will encourage the reviewer to raise their score. We will be doing our best to continue to answer any outstanding issues during the discussion period.

---

> ### Author Response · Authors · 2021-08-10
> **Results on MobileNetV2 (MBV2) on ImageNet (Updated!)**
>
> ---
> Edit on 08/28/21:
> * We updated our results to include the latest set of comparisons.
> * Since the initial response, we ran all of the baseline comparison methods in a one-shot compression experiment _with retraining_ akin of Section 3.1
> * Consequently, we have removed the prune-only results as they were not as insightful for our tested compression ratios.
> * We replaced the compress-only results with a full suite of comparisons in the one-shot compress+retrain scenario
> * We also removed comparison to the weight pruning method as it is hard to accurately compare weight pruning and low-rank compression.
> ---
>
> Below, we are providing our results for MBV2 on ImageNet.
>
>
> ### Hyperparamters and experiment setup
>
> For our experiments, we follow the one-shot setup of Section 3.1 (train+compress+retrain).
>
> For training, we use the pre-trained [MBV2 torchvision model](https://arxiv.org/pdf/1801.04381.pdf).
>
> During compression, we try various compression ratios as reported.
>
> For retraining, we follow the hyperparameter setup in the [MBV2 paper](https://arxiv.org/pdf/1801.04381.pdf) except that we lower the initial learning rate and we retrain for fewer epochs compared to the training to ensure that we can report our results in a timely manner during the discussion period.
>
> Retraining hyperparameters:
> * Optimizer: RMSProb with momentum 0.9, alpha 0.9, and weight decay 0.00004
> * 100 epochs
> * Learning rate schedule: 1.85e-6 initial learning rate, 0.98 learning rate decay per epoch
> * batch size 768 trained over 8 GPUs
>
> ### Baseline accuracy
>
> The baseline model achieves Top1- and Top5-accuracy of  71.85% and 90.33% accuracy, respectively.
>
> ### Comparisons
>
> We could not find any other either structured pruning papers or tensor decomposition papers that previously reported results for MBV2 (ImageNet). So we ran our tensor decomposition baseline comparison methods from Section 3.1 to be able to compare against other methods.
>
> Note that we did not have the computational resources to also run our baseline filter pruning comparisons yet (FT and PFP) and we wanted to focus on the most relevant baseline comparisons (tensor decomposition methods). But we will make sure to include them as soon as they are ready.
>
> ### Compress+retrain results
>
> Below, we present results from our one-shot compress+retrain experiment on MBV2 (ImageNet). We provide multiple tables in each of which we provide the difference in Top1 and Top5 test accuracy compared to the uncompressed model. We also provide the overall relative reduction in terms of parameters and FLOPs, denoted by CR-P and CR-F.
>
> Each table refers to a different parameter compression ratio (CR-P) we tested. Empty lines indicate that the comparison method was not (yet) tested for the specific CR-P.
>
>
>
> ___
> **CR-P $\approx$ 20%:**
>
> | | Top1 Accuracy Diff | Top5 Accuracy Diff | CR-P | CR-F |
> |---|---|---|---|---|
> | ALDS | -1.17% | -0.37% | 19.9% | 5.1% |
> | PCA | -2.87% | -0.55% | 20.9% | 3.3% |
> | SVD-Energy | -1.27% | -0.56% | 20.0% | 8.6% |
> | SVD | -3.65% | -2.07% | 20.0% | 32.0% |
> | L-Rank | -19.08% | -13.40% | 20.0% | 62.0% |
>
>  ___
>
> **CR-P $\approx$ 33%:**
>
> | | Top1 Accuracy Diff | Top5 Accuracy Diff | CR-P | CR-F |
> |---|---|---|---|---|
> | ALDS | -1.53% | -0.72% | 33.0% | 11.0% |
> | PCA | -4.29% | -2.37% | 32.6% | 11.9% |
> | SVD-Energy | -2.50% | -1.45% | 32.7% | 20.8% |
> | SVD | -5.41% | -3.09% | 32.7% | 43.1% |
> | L-Rank | -35.54% | -28.40% | 32.7% | 75.4% |
> ___
>
> **CR-P $\approx$ 43%:**
>
> | | Top1 Accuracy Diff | Top5 Accuracy Diff | CR-P | CR-F |
> |---|---|---|---|---|
> | ALDS | -3.77% | -2.16% | 43.5% | 18.3% |
> | PCA | -9.02% | -5.48% | 43.2% | 24.2% |
> | SVD-Energy | -4.05% | -2.42% | 43.4% | 33.1% |
> | SVD | -8.08% | -4.94% | 43.4% | 53.1% |
> | L-Rank | -66.04% | -72.89% | 43.4% | 85.2% |
>   ___
>
> **CR-P $\approx$ 52%:**
>
> | | Top1 Accuracy Diff | Top5 Accuracy Diff | CR-P | CR-F |
> |---|---|---|---|---|
> | ALDS | -6.45% | -3.68% | 52.4% | 31.0% |
> | PCA | N/A
> | SVD-Energy | N/A |
> | SVD | -10.62% | -6.60% | 52.4% | 60.3% |
> | L-Rank | N/A |
>   ___
>
>
> ### Interpretation of results
>
> As with other networks, we can see that ALDS performs favorably both in terms of parameter reduction and FLOP reduction compared to the other methods on MBV2 (ImageNet).
>
> Interestingly enough, we can still get a significant reduction in parameters (CR-P), however, it seems that reducing the number of FLOPs is harder. Moreover, the reduction in FLOPs (CR-F) does not correlate as well with the reduction in parameters (CR-P) as for other networks. This may be explained by the fact that MBV2 is an architecture that was specifically designed and optimized with a reduction in FLOPs in mind.
>
> Nevertheless, it is encouraging to see that ALDS can further reduce both the parameter count and FLOP count and that ALDS outperforms other methods we compare to.
>
> For example, for a CR-P of 20%, 33%, 43%, and 52% ALDS exhibits a difference in Top1-accuracy of -1.17%, -1.53%, -3.77% and -6.45%, respectively, compared to -1.27%, -2.50%, -4.05%, and -10.62%, respectively, for the best competing baselines in each scenario. Note that a more negative difference in accuracy indicates worse performance as it indicates a bigger drop in accuracy compared to the original, unpruned network.

---

> ### Author Response · Authors · 2021-08-28
> **We have addressed your concerns + updated results**
>
> Dear Reviewer AgkJ,
>
> Thank you again for your insightful review of our paper. It has already helped us to significantly improve our manuscript and we are looking forward to further improving our paper in case you have any remaining concerns.
>
> Since the discussion period is coming to an end, we would like to reach out to you at this point in the review process and kindly ask whether there are any additional concerns or issues that you would like to bring to our attention.
>
> We are also happy to announce that we significantly updated our results on the MBV2 architecture for ImageNet  (**see [here](https://openreview.net/forum?id=BvJkwMhyInm&noteId=28Upw1tqCj)**).
>
> As far as we can tell we are actually the first tensor decomposition technique that tries to compress MBV2 and one of the first tensor decomposition techniques aiming to compress an architecture that has already been designed with significant FLOP reductions in mind.
>
> We would like to emphasize that this is not standard practice in the pruning literature as such efficient architectures are naturally less compressible compared to standard architectures and thus benchmark/comparison results might be distorted or less indicative of the true performance on a more generic architecture.
>
> Overall, however, we are truly grateful for bringing such an experiment to our attention as it will significantly strengthen the final paper.
>
> From analyzing your initial review and our responses so far, it seems that we have addressed all your concerns to a great extent. We cannot identify any lingering issues that would prevent you from raising your score and thus voting for the acceptance of our paper as you indicated in your initial review.
>
> We look forward to hearing back from you and addressing any remaining concerns you might have. Thank you!

---

### Official Review · Reviewer_auNv · 2021-07-19

**Rating:** 5
**Confidence:** 4

**Summary:**

The paper proposes a new layer-wise compression technique to compress neural networks. The new compression method applies channel slicing method to divide convolution layer into k subsets, followed by using SVD method (similar to Denton et al. (2014) method) to compress each layer. The paper further proposes an automatic layer-wise decomposition selector (ALDS) to compress the whole networks with the proposed techniques by optimizing global network compression (equation (8)). Experiment results on Cifar10, ImageNet and VOC compared to the other ow rank method show that the proposed ALDS has a better prediction performance at the considered compression ratio.

**Limitations And Societal Impact:**

Not aware of any negative social impact.

**Main Review:**

This paper is an extension work of Denton et al. (2014) method, which involves new proposed techniques, including channel slicing, SVD approximation, ALDS algorithm, and iterative fine tuning. The algorithms seem to be some adhoc changes compared to Denton et al. (2014) method, particularly when the slicing method is just random slicing. The experiment results, without comparing to the benchmark tensor compression method, might be difficult to convince the effectiveness of the algorithm.

Some questions to the methods used in this paper are
- Channel slicing methods (line 88-90) in this paper seem to be a random partition of slice into k groups. How is the performance of channel slicing method compared to clustering algorithm based slicing method in model compression? How to select k when applying the compression method in practice?
- The paper conduct experiment on variance of datasets using bench mark backbone models. However, convolution layer is a tensor and known to be redundant in parameters, and used to be compressed by low rank methods, such low rank tensor methods (Kim et al, 2015, Novikov et al. 2015). Slicing the matrix into parts leveraged the tensor structure of convolutions layers. Without comparing the algorithm with other tensor compression method, it might be difficult to tell the performance gains.
- Could the author helps to clarify more details about how to find the optimal subspaces k based on layer budget b (Algorithm 1, line 7)?
- In line 81, the paper claims after SVD decomposition of convolution layer, one convolution layer becomes two small convolution layers. Are there any non-linear transformation considered between the two convolution layers?

**Time Spent Reviewing:**

2

---

> ### Author Response · Authors · 2021-08-10
> **Response to Reviewer auNv - Part 1/2: Opening Statement, Channel Slicing, Comparison Methods**
>
> ## Thank you
>
> We thank the reviewer for the comments and we are looking forward to further engaging with the reviewer during the open discussion period. To kick off the discussion, we are providing the answers to the issues and questions you are raised in your initial review.
>
> We hope that the reviewer will consider changing their score given our detailed response. Please do not hesitate to reach out if you require further clarification.
>
> ## Channel slicing and selection of $k$
>
> ### Issue raised by the reviewer
>
> _Channel slicing methods (line 88-90) in this paper seem to be a random partition of slice into k groups. How is the performance of channel slicing method compared to clustering algorithm based slicing method in model compression? How to select k when applying the compression method in practice?_
>
> ### Our response
>
> #### **Relevant section in existing manuscript**
>
> We would encourage the reviewer to re-read the corresponding paragraph in the paper (lines 82-92) that the reviewer referenced. Specifically, the paragraph states that *prior* methods considered random partitioning of input channels into k groups but that this may lead to significant slow-downs during runtime (lines 83-88). Thereafter, we specifically state (lines 88-92) that we devise a channel slicing technique that is not random and is indeed efficiently implementable in practice.
>
> Below, we quote lines 88-92 from the paper for your reference:
>
> _We instead opted for a simple clustering method, namely channel slicing, where we simply divide the c input channels of the layer into k subsets each containing at most dc=ke consecutive input channels. Unlike other methods, channel slicing is efficiently implementable, e.g., as grouped convolutions in PyTorch (Paszke et al., 2017), and ensures practical speed-ups subsequent to compressing the network._
>
> To summarize, channel slicing is not a random operation but rather a "grouped convolutions". Further details about are clustering techniques and our channel slicing are also provided in Section A.2 of the appendix.
>
> #### **Channel slicing vs. clustering**
>
> We also conducted experiments to compare this channel slicing method with a clustering algorithm method in model compression. To this end, we would like to point the reviewer to our ablation study (Figure 7 in Section 3.3 and a more complete discussion in Section C.3 of the appendix). Here, we compare our subspace clustering (channel slicing) to the clustering technique of Maalouf et al., 2021, which clusters the matrix columns using projective clustering (we call this method `Messi` following Maalouf et al.).
>
> We would like to specifically point the reviewer to the comparison between `ALDS-Simple` and `Messi`. Both of these methods compress the network in the exact same manner _except_ for the clustering technique used. `ALDS-Simple` uses our channel slicing; `Messi` uses projective clustering as proposed by Maalouf et al., 2021.
>
> Given that `Messi`is a more general clustering technique, we may expect that `Messi` improves the performance over ALDS-Simple. However, as shown in Figure 7 the increase is very slight and the difference is essentially negligible. Furthermore, we can clearly see that our proposed overall scheme (`ALDS`) outperforms all of these variants including `Messi`. This is due to the fact that it applies a search in the optimal values of k and j for each layer with respect to the global error, and since the inner method “channel slicing” is simple and fast, we can do such a search in a suitable amount of time.
>
> Together with the computational disadvantages of Messi-like clustering methods (unstructured, NP-hard; see Section 2.1) ALDS-based simple channel slicing is therefore the preferred choice in our context.
>
> #### **Choice of $k$**
>
> Moreover, to address the last point of your question we would like to highlight that $k$ is automatically chosen for each layer following the optimization procedure outlined in Algorithm 1, Section 2.3. Specifically, in line 7 of Algorithm 1 we optimize for $k^\ell$ in each layer. We also encourage the reviewer (if interested) to read Section 2.3, Section A.4, and our response to *Issue 2* raised by Reviewer piFe, where we provide additional details regarding Algorithm 1.
>
> ## Comparison to tensor decomposition methods
>
> ### Issue raised by the reviewer
>
> _The paper conduct experiment on variance of datasets using bench mark backbone models. However, convolution layer is a tensor and known to be redundant in parameters, and used to be compressed by low rank methods, such low rank tensor methods (Kim et al, 2015, Novikov et al. 2015). Slicing the matrix into parts leveraged the tensor structure of convolutions layers. Without comparing the algorithm with other tensor compression method, it might be difficult to tell the performance gains._
>
> ### Our response
>
> We thank the reviewer for raising this important issue and would like to further clarify how our method fits into the broader context of tensor decomposition methods.
>
> #### **Contextualization within related work**
> To this end, we would like to point the reviewer to our discussion on related work (Section 4, paragraph “Low-rank compression (local step)”). There, we provide a thorough overview of related work and different approaches to low-rank decomposition.
>
> Notably, one of the most prevalent approaches to tensor decomposition entails folding the tensor into a matrix, decomposing it via SVD, and unfolding the pair of matrices into a pair of tensors. This approach was used in various papers in recent years (among others by Denton et al. (2014); Sainath et al. (2013); Tukan et al. (2020); Xue et al. (2013); Yu et al. (2017) as cited in our paper).
>
> Other approaches to tensor decomposition include decomposing the tensor directly without taking the step of folding the tensor into a matrix first and we also cite the relevant work there.
>
> #### **Experimental comparisons as requested by the reviewer are already included**
>
> We also incorporated a lot of comparison methods into our benchmark experiments in Section 3.2 (Table 2).
>
> Specifically, the following comparison methods referenced in Table 2 are tensor decomposition methods that do not rely on SVD as sub-procedure but rather directly decompose the tensor:
>
> * `MUSCO`, Gusak et al., 2019: using Tucker and HO-SVD for tensor decomposition
> * `PCA`, Zhang et al, 2015a: non-linear PCA with custom tensor decomposition
> * `Expand`, Jaderberg et al., 2014: custom, bi-convex optimization for tensor decomposition
> * `Tucker`, Kim et al., 2015a: Tucker decomposition
> * `Coordinate`, Wen et al., 2017: custom approach via “force” regularization
>
> From the two references mentioned by the reviewer, Kim et al., 2015a, is already incorporated into the experimental results. We checked Novikov et al., 2015, however, they do not seem to provide results pertaining to compressing the entire network and rather focus on compressing individual layers. On the other hand, our approach is focused on compressing the entire network.
>
> We believe our current experiments thus already provide sufficient evidence on the effectiveness of our method compared to tensor decomposition techniques.
>
> If, however, the reviewer would like to see additional comparisons, we would be more than happy to incorporate them. If so, please do reply to us in a timely manner so that we can incorporate additional comparisons as requested.

---

> ### Author Response · Authors · 2021-08-10
> **Response to Reviewer auNv - Part 2/2: Clarifications, Final Remarks**
>
> ## Clarification with regards to Algorithm 1
>
> ### Issue raised by the reviewer
>
> _Could the author helps to clarify more details about how to find the optimal subspaces k based on layer budget b (Algorithm 1, line 7)?_
>
> ### Our response
>
> Thank you for inquiring about additional details regarding our Algorithm 1 (ALDS).
>
> #### **Relevant sections in current manuscript**
> For this, we would like to refer the reviewer to Section A.4., lines 765-769, in the appendix. There, we provide additional clarification with regards to line 7 of Algorithm 1.
>
> #### **Further clarifications**
> We would also like to point the reviewer to our response of *Issue 2* to Reviewer piFe which goes into depth regarding Algorithm 1. For your convenience, we copy/paste the most relevant parts of our answer to Reviewer piFe below:
>
> #### $OptimalSubspaces(b^\ell)$
>
> First, we note that for this step we proceed on a per-layer basis. Here, for each layer we are given specific values of $k^\ell$ and $j^\ell$, which implies that we are given a budget $b^\ell$ for every layer $\ell\in[L]$.
> Subsequently, we re-assign the number of subspaces $k^\ell$ and their ranks $j^\ell$ for each layer $\ell$ as follow:
> We iterate through the finite set of possible values for $k^\ell$, for every such value $k^\ell$ we pick its corresponding $j^\ell$ such that the total size (number of parameters) of this layer is (approximately) the given budget $b^\ell$.
> Now, for every pair of candidates $k^\ell$ and $j^\ell$ we compute the relative error on this layer that is caused after compression with respect to these values.
> Finally, we choose the combination of $j^\ell$, $k^\ell$ that minimizes the relative error for the current layer budget $b^\ell$. We then discard the values found for $j^1, \ldots, j^L$ and re-optimize them in the next iteration of $OptimalRanks$.
>
> ## Two small convolutional layers
>
> ### Issue raised by the reviewer
>
> _In line 81, the paper claims after SVD decomposition of convolution layer, one convolution layer becomes two small convolution layers. Are there any non-linear transformation considered between the two convolution layers?_
>
> ### Our response
>
> Thank you for this interesting suggestion. We do not consider any non-linearities between the two resulting convolutional layers. The main purpose of splitting the original layer into two layers after the low-rank decomposition is to ensure a dense reparameterization that requires less FLOPs overall compared to the original layer. This is standard across most (or all) the compression papers that leverage low-rank decomposition techniques as far as we know.
>
> However, we definitely agree that this could be a very relevant avenue for future work where we could investigate if we can further boost the approximation accuracy for a given low-rank compression using some form of additional non-linearities. One potential downside of this idea, however, could be that the additional non-linearity slows down the runtime. All of this would require some deeper investigation, which would be beyond the scope of this paper.
>
> ## Final remarks
>
> We thank the reviewer again for their insightful comments and we are looking forward to continuing the discussion. Hopefully, the reviewer will consider raising their score given that we have addressed all their feedback.

---

> ### Author Response · Authors · 2021-08-28
> **We have addressed all your concerns**
>
> Dear Reviewer auNv,
>
> As the discussion period is coming to an end, we would like to reach out again and inquire about any potential lingering concerns you might have.
>
> We just re-assessed your initial review and our response.
>
> As apparent from our initial response, most issues seem to be related to some points of confusion that we were able to solve by pointing the reviewer to the respective sections in the manuscript. We also tried to carefully assess if our current manuscript might lack clarity in certain aspects and, if so, provide additional details in our response (e.g. clarification around Algorithm 1, line 7).
>
> Consequently, we believe that we have therefore been able to address all your initial concerns and we would like to kindly ask the reviewer to re-assess their evaluation and score of our paper as we cannot identify any more lingering issues that could potentially preclude the paper from acceptance.
>
> We are more than happy to continue addressing any issues you might have and to engage further in a discussion in order to increase your score. If we receive a response from you, we will try to respond as quickly as possible in order to not delay the decision process for acceptance.
>
> Thank you!

---

### Official Review · Reviewer_piFe · 2021-07-20

**Rating:** 4
**Confidence:** 4

**Summary:**

The paper proposes a one-shot approach to compress already trained neural networks using low-rank compression. The authors propose a new scheme of applying low-rank: instead of applying it directly to a layer, authors suggest partitioning the filters into K blocks, and apply low-rank to each of these blocks separately. Haven defined this scheme, authors then propose a one-shot alternating-optimization algorithm to select ranks and to partition matrix columns/filters into K blocks. Once ranks and partitions are determined, the matrices are decomposed and finetuned. Optionally, this entire compress=>finetune process can be repeated multiple times. Authors present numerical results where the suggested approach almost always wins, however some of the comparisons are questionable. In terms of the algorithm itself, the details of alt.opt are missing, as well as theoretical analysis of the proposed compression scheme is missing.

**Ethical Concerns:**

I do not see any ethical concerns associated with this work.

**Limitations And Societal Impact:**

I do not believe that the paper has a societal impact that should be addressed.

**Main Review:**

The paper is well written, and the method's main idea is presented well; the literature review is up to date. The authors present an impressive amount of experimental validations, however, there are several limitations of this work stemming from its main contributions. To remind the reader, there are two main contributions of this paper:
   1) a new low-rank compression scheme
   2) a simple one-shot alternating optimization
which in combination allows to achieve a quick and accurate (based on the results) compression. Unfortunately, both of these contributions lack a substantial analysis and might actually lead to a subpar (theoretically) performance when compared to a regular low-rank compression. Let me explain my analysis next.

**Issue 1** Analysis of a new compression scheme. The authors, in the "multiple subspaces" paragraph (line 82-92) propose to partition (slice) the columns of the matrix and apply j-rank compression to a clustered subspaces and then combine them. For instance, assuming $k=2$ partitioning, if we have a matrix of size $m\times n$, authors propose to split it into $m \times a$ and $m \times b$ (with $a+b=n$) matrices and apply $j$-rank approximation to both matrices. This allows to achieve a compression ratio of $\frac{mn}{2mj + 2j(a+b)}$. In a sense, this suggested scheme is equivalent (in required parameter size) to a $kj$-rank decomposition ($2j$ in our example). However, if we were to apply $kj$-rank-decomposition immediately, by Eckart-Young-Minsky theorem, we would have a minimal approximation error out of all $kj$-rank decompositions available.

Here is an important question, whether the authors' suggested scheme achieves better than regular SVD approximation when keeping all parameters ($k,j$) fixed? The authors' bound derived by the same Eckart-Young-Minsky theorem does not answer this question, thus I have written a MATLAB script to empirically quantify the approximation errors achieved by the authors' method and by regular SVD.  For a randomly sampled matrices (from a Gaussian), with $k=2$ partitions, $kj$-rank SVD achieves a better approximation error than the author's suggested scheme. This might suggest that author's scheme is inherently less powerful. Certainly, more investigation is needed. (I'm attaching the script below)


**Issue 2** The proposed Algorithm 1 lacks details. The algorithm 1, is being dumped on the reader without a sufficient number of details. First of all, how do OpimalRanks and OptimalSubspaces work? Since details are lacking, it is hard to understand whether this consists of convergent iterations (i.e., each step reduces the objective?). Supplemental materials lack the necessary details as well.


Beyond these two issues, there are other important ones:
**Issue 3** Fairness of some of the experimental evaluations. In section 3.1 authors compare to a bunch of baselines, out of which comparison to L-rank seems unfair. The L-rank method requires running an entire pipeline of training, and not of the "one-shot" compression method. However, authors seem to run in in a "one-shot" setting, in which the method unsurprisingly yields bad results. This is not an apples-to-apples comparison.  At the same time, in table 2,  in the apples-to-apples setting, the quoted L-rank result is on par with the author's method.

**Issue 4** Relevance of theorem 1? Authors spend sufficient time on stating and proving the theorem 1, however, its results are not being used later in the algorithm (at least the details are missing).


**Overall**, my current evaluation is between "reject" to "borderline; tending to reject".

Below is the Matlab script to quickly assess the proposed compression scheme.  To verify my experimental conclusions mentioned above (that it has worse approximation error when compared to regular SVD when evaluated on randomly sampled $100\times50$ matrix  with $k=2$ and $j=20$) you can invoke the function as `analyze_proposed_scheme(randn(100,50), 20)`. These findings urge a more in-depth analysis of the scheme's performance.
```matlab
function analyze_proposed_scheme(X, r)
    m = size(X,2);
    authors_min_error = Inf;
    split_at = Inf;
    for s=r:m-r
        a = X(:,1:s);
        [u1,s1,v1] = svd(a);

        b = X(:,s+1:end);
        [u2,s2,v2] = svd(b);
        scheme_approx = [u1(:,1:r)*s1(1:r,1:r)*v1(:,1:r)' u2(:,1:r)*s2(1:r,1:r)*v2(:,1:r)'];
        curr_err = norm(X-scheme_approx, 'fro');
        if curr_err < authors_min_error
            authors_min_error = curr_err;
            split_at =s;
        end
    end
    authors_min_error

    [u,s,v] = svd(X);
    % the proposed scheme is equivalent in parameter size to kj-rank
    % decomposition, here we have k=2, and j=r
    X_svd = u(:,1:2*r)*s(1:2*r,1:2*r)*v(:,1:2*r)';

    svd_error = norm(X-X_svd, 'fro')
end
```

**Time Spent Reviewing:**

4

---

> ### Author Response · Authors · 2021-08-10
> **Response to Reviewer piFe - Part 1/2: Opening Statement, Issue 1**
>
> ## Thank you
>
> We would like to thank the reviewer for their positive evaluation of our work and their constructive criticism to help improve our manuscript.
>
> Below we discuss the issues raised by the reviewer in more detail. We hope we have sufficiently addressed all of your concerns and we would be more than happy to continue to engage during the discussion period with regards to any remaining concerns you might have about our manuscript.
>
> ## Issue 1
>
> We thank the reviewer for the careful reading and for proposing a simple script to practically benchmark our approach. However, there seems to be a simple (yet very important) point that may have been missed in the review which caused the script to be incorrect. Specifically, the script compares our approach to the standard SVD factorization while using different sizes:
>
> #### **Comparison of sizes**
>
> For an $m \times n$ real matrix. When we apply our compression technique with, e.g., $k$ clusters and $j$ as the low rank, the number of parameters that are used to define our low-rank points (new layers) is $kjm+ nj$ (*not* $kjm + kjn$ ). This is due to the following. We first cluster the $n$ columns into $k$ subsets.
> Now for each subset, we compute its own $j$-dimensional subspace that requires $jm$ coordinates to be represented in $R^m$, i.e., $kmj$ parameters to represent the $k$ subspaces for the $k$ subsets.
> However, we still have only $n$ points (columns), and each point is now projected on its relevant rank-$j$ subspace, i.e., we need only $j$ coordinates to represent each point, and only $jn$ coordinates to represent the whole set of projected points.
> Hence, $kjm+ nj$ parameters in total represent the subspaces and the projected points.
>
> This is also stated on lines 106, 107 of our manuscript and can also be inferred from Figure 3. Please let us know if this was not clear from the current version of the manuscript and we would be happy to further clarify it in an updated version once we are allowed to change the manuscript again.
>
> #### **Changes to evaluation script**
>
> Moreover, we also adapted your proposed testing script to account for a fair comparison in terms of the number of parameters. Specifically, for your example when $k=2$, and $n=a+b$, the compression ratio is: $nm/(2jm + nj)$, and not $nm/(2jm + 2j(a+b))$, and since the number of parameters that is required to represent the optimal $kj$ subspace (as suggested by the reviewer) is $kj(n+m)>kjm+ nj$, the comparison is unfair as we can use a larger number of the parameter in our technique to achieve better results.
> The right way to do such a comparison is to pick the value $j_2$, where $j_2(n+m) \approx kjm+ nj$, i.e., $j_2 \approx (kjm+nj)/(n+m)$.
>
> #### **Updated evaluation script**
>
> Below, you may find an updated version of the Matlab script (we compare $k=2$ and $k=1$ as suggested by the reviewer), but we define the right $j$ for SVD with $k=1$ to obtain the same number of parameters as in the case of  $k=2$.
> The results clearly show that most of the time $k=2$  is better (in our 20 trials it was always better) for the example provided by the reviewer, indeed this may not always be the case, and trying more $k$ values could bring better results.
>
> ```
> function analyze_proposed_scheme(X, r)
>     n = size(X,2);
>     m = size(X,1);
>     k = 2;
>     authors_min_error = Inf;
>     split_at = Inf;
>     for s=r:n-r
>         a = X(:,1:s);
>         [u1,s1,v1] = svd(a);
>
>         b = X(:,s+1:end);
>         [u2,s2,v2] = svd(b);
>         scheme_approx = [u1(:,1:r)*s1(1:r,1:r)*v1(:,1:r)' u2(:,1:r)*s2(1:r,1:r)*v2(:,1:r)'];
>         curr_err = norm(X-scheme_approx, 'fro');
>         if curr_err < authors_min_error
>             authors_min_error = curr_err;
>             split_at =s;
>         end
>     end
>     authors_min_error
>
>     [u,s,v] = svd(X);
>     r_for_svd = (k*r*m+ n*r)/(m+n)
>     % the proposed scheme is equivalent in parameter size to r_for_svd-rank
>     X_svd = u(:,1:r_for_svd)*s(1:r_for_svd,1:r_for_svd)*v(:,1:r_for_svd)';
>
>     svd_error = norm(X-X_svd, 'fro')
> end
> ```
>
> Use the call (as in your example): `analyze_proposed_scheme(randn(100,50), 20)`. We used GNU Octave to run your script and our modified script.
>
> #### **Further insights about our approach versus SVD**
>
> Furthermore, we note that our approach is a generalization of SVD ($k=1$ always) to any $k \geq 1$ and as such is always at least as good as “vanilla” SVD. The motivation of our work is automatically determining the pair $k,j$ for every layer, such that the error is minimized while the total number of parameters (i..e the desired compression ratio) is fixed. In some layers, we may have $k=1$, but this is not a must, in fact, our experiments show that most of the times we obtained $k \geq 2$ for most layers.
>
> #### **Ablation study for comparison**
>
> Finally, we would like to point the reviewer to our ablation study where we tested various features of our proposed low-rank compression scheme (see Section 3.3 and Section C.3 for an extended discussion). Specifically, we would like to point out the difference in performance between `ALDS` , `ALDS-Error`, and `SVD-Error`. `SVD-Error` hereby follows the exact same algorithm as ALDS (Algorithm 1) with the exception that $k=1$ is fixed for all layers. We also tested a variant denoted by `ALDS-Error` that fixes $k=3$.
> From these comparisons we can draw two conclusions. First, we can see that `ALDS-Error` improves over `SVD-Error` implying that $k>1$ is oftentimes beneficial. Moreover, we observe that `ALDS` improves over `ALDS-Error` implying that varying $k$ is also beneficial.
>
> #### **Conclusion of Issue 1**
>
> We hope this answer solves the reviewers' concerns.

---

> > ### Comment · Reviewer_piFe · 2021-08-21
> > **Thanks for clarifying Issue 1**
> >
> > Dear authors, thanks a lot for the detailed clarification. I now see that your scheme indeed has $kjm+jn$ parameters and not $kjm+kjn$. I stand corrected on this matter, however, this does not entirely solve the issue 1: the analysis of the proposed scheme. Experimentally, I was able to found cases where svd still wins even with the updated script: for example, if you try `analyze_proposed_scheme(randn(100,100), 18)` several times, you might encounter such cases as well. Therefore, there is still a question of  "Whether the ALDS scheme always better than SVD with the equivalent number of parameters?" and most importantly, "whether the algorithm to obtain the ALDS scheme gives a globally optimal ALDS scheme that cannot be further improved".
> >
> > Let me clarify the last question. The solution to getting the "ALDS" scheme is a list of specific actions: split, decompose, combine. Let's call the resulting decomposed matrix $\hat{X}$, and in the paper you bound the approximation error $||X-\hat{X}||$ wrt $\hat{X}$. The question is whether $\hat{X}$ is the most optimal decomposition (in $\ell_2$ sense) wrt possible ALDS decompositions? If yes, then all questions are solved, and by the result of "split, decompose, combine" steps we get the globally optimal answer. However, practical evaluations of this scheme (as well as the absence of guarantees about optimality) clearly tell that there are ways of getting even better ALDS decompositions.

---

> > ### Author Response · Authors · 2021-08-21
> > **Thank You and Additional Remarks - Issue 1**
> >
> > Thank you so much for your reply. To further address your concerns regarding Issue 1, we would like to note the following:
> >
> > ### **1.**
> >
> > ALDS finds for each layer the optimal decomposition, such that the overall error across all layers is minimized, while simultaneously achieving a specific overall compression ratio (for the whole network). Note that we search over all layers and we cannot obtain a global optimum across all layers since this is a combinatorial problem (the assignment of the optimal per-layer compression ratio is combinatorial). However, each iteration of ALDS minimizes the per-layer error (i.e. finds the optimal decomposition) resulting in a better local optimum.
> >
> >
> > ### **2.**
> >
> > **ALDS may find that SVD ($k=1$) is the optimal solution for some layers** in some scenarios as we have already mentioned in our previous response.
> >
> >
> > ### **3.**
> >
> > The analysis script that you have been running (the Matlab code) compares **two possible outcomes** of our decomposition scheme **for one layer**, i.e. $k=2$ and $k=1$ (classic SVD). Indeed in some cases and for some datasets using **ALDS decomposition with $k=1$ (classic SVD) will give a better approximation error** than $k=2$.
> >
> > Additionally, we would like to note the following:
> >
> > 1. Note that in your script you compare $k=1$ and $k=2$. Sometimes, $k=1$ is indeed better than $k=2$ but that does not contradict our result since we optimize for the best $k$. Moreover, it might be that there is another $k>2$ that might be optimal.
> >
> > 2. ALDS checks all possible values of $k$ for each layer and picks the best for each layer, which might be $k=1$ (SVD) for some layers. This is line 7 of Algorithm 1 ($OptimalSubspaces$). This is related to the distribution of the data and the goal of ALDS is to find which factorization (i.e. which $k$ and $j$ in each layer) fits the data distribution the most. If, for example, a situation occurs where $k=1$ is better than $k=2$ (like sometimes in your script) then ALDS will pick $k=1$ according to line 7 of Algorithm 1 and it will reduce to SVD. Note that this is also reflected in Theorem 1 from which we get the approximation error for any desired $k$. **We will ensure this part is more accessible in the final manuscript (see Issues 2 and 4 as well)**.
> >
> > 3. The comparison in the Matlab script is done for one layer. In real scenarios ALDS performs on multiple layers.
> >
> > 4. If $k=1$ there are no “split, combine” steps but only the “decompose” step (we “fake-split” into one set).

---

> ### Author Response · Authors · 2021-08-10
> **Response to Reviewer piFe - Part 2/2: Issue 2, Issue 3, Issue 4, Concluding Remarks**
>
> ## Issue 2
>
> #### **Reference to algorithmic details in current manuscript**
>
> We thank the reviewer for pointing this out. In the current manuscript, we discuss the details of Algorithm 1 at a high level in Section 2.3 and discuss additional details of our method in Section A of the appendix (Section A.4 is hereby probably of most interest to the reviewer).
>
> Below, we include additional details and clarification regarding Algorithm 1, which we will incorporate into our final manuscript. Most details are already discussed in Section A.4 but we hope our answer will make the details of ALDS even more clear and accessible.
>
> #### **Overview of Algorithm 1**
>
> At a high-level, Algorithm 1 aims to find a local optimum for the optimization procedure described in Equation (8). We hereby iteratively optimize for $k^1, \ldots, k^L$ and $j^1, \ldots, j^L$. The step where we fix the set of $k$’s and optimize for the set of $j$’s is line 4, whereas in line 7 we fix the layer budget and optimize for the set of $k$’s. At each step the objective is minimized. Thus for a fixed seed, ALDS converges to a local optimum of (8). We then repeat the entire procedure multiple times with different random seeds to improve the quality of the local optimum.
>
> Next we provide additional details with regards to $OptimalRanks$ and $OptimalSubspaces$.
>
> #### $OptimalRanks(\mathbf{CR}, k^1, \ldots, k^L)$
>
> At this step we are given a guess for the optimal values of $k^1,\dots, k^L$, and our goal is to compute the values $j^1,\dots, j^L$ that minimize the objective function described in Equation (8), i.e., the maximum error $\max_{\ell \in [L]} \varepsilon^\ell$, while achieving the desired global compression ratio $\mathbf{CR}$.
>
> To find the optimal solution, we note the following. Recall that $k$’s are fixed.
>
>
> 1. **The maximum error is minimized exactly when all errors are equal.** To see that this is indeed the case we can proceed by contradiction. Suppose we found an optimal solution where all errors are not equal. Then we could use some of our compression budget to add more parameters to the layer with the maximum error while removing the same amount of parameters from the layer with minimum error. Since adding more parameters improves the error we just lowered the maximum error by adding more parameters to the layer with the maximum error. Hence, this leads to a contradiction proving our initial statement.
>
> 2. **A given constant error across layers corresponds to a fixed compression ratio.** This should be very straightforward to see. Specifically, for a given layer error we can find the corresponding rank and the rank implies how many parameters the compressed layer will have. This then implies a fixed compression ratio. Moreover, note that this relation is monotonic.
>
> Both (1.) and (2.) together imply that we can use a binary search or some other root finding algorithm to determine the corresponding constant error for a desired compression ratio. The solution of our binary search will then be the corresponding set of $j$’s (ranks) for each layer that minimizes the maximum error for a desired compression ratio and given set of $k$’s (recall that we optimize for $k$’s separately).
>
> #### $OptimalSubspaces(b^\ell)$
>
> First, we note that for this step we proceed on a per-layer basis. Here, for each layer we are given specific values of $k^\ell$ and $j^\ell$, which implies that we are given a budget $b^\ell$ for every layer $\ell\in[L]$.
> Subsequently, we re-assign the number of subspaces $k^\ell$ and their ranks $j^\ell$ for each layer $\ell$ as follow:
> We iterate through the finite set of possible values for $k^\ell$, for every such value $k^\ell$ we pick its corresponding $j^\ell$ such that the total size (number of parameters) of this layer is (approximately) the given budget $b^\ell$.
> Now, for every pair of candidates $k^\ell$ and $j^\ell$ we compute the relative error on this layer that is caused after compression with respect to these values. The relative error is hereby computed according to Theorem 1.
> Finally, we choose the combination of $j^\ell$, $k^\ell$ that minimizes the relative error for the current layer budget $b^\ell$. We then discard the values found for $j^1, \ldots, j^L$ and re-optimize them in the next iteration of ${OptimalRanks}$.
>
> Note that for ${OptimalSubspaces}$ there is no monotonic relation between the value of $k^\ell$ and the corresponding error like there is between the value of $j^\ell$ and the error. Hence, we proceed on a per-layer basis where we keep the per-layer budget constant during ${OptimalSubspaces}$ as described above.
>
> #### **Optimality**
>
> From the details of the two steps, it should be very clear that the cost is decreasing at each step in the optimization procedure and we can thus conclude that for each random seed Algorithm 1 converges to a local optimum (at which point the cost will be non-increasing).
>
> #### **Additional Remarks**
>
> Note that above for ${OptimalRanks}$ we assumed that the errors ($\varepsilon^\ell$’s) are continuous but they are actually discrete given that they are a function of the rank which is discrete. However, as long as we can ensure that the objective decreases at every iteration we can still reach a local minimum.
>
> Alternatively, we can solve the continuous relaxation of the above problem and use a [randomized rounding](https://en.wikipedia.org/wiki/Randomized_rounding) approach to get an approximately optimal solution.
>
> In practice, however, we found that it is not necessary to add this additional complication step since it is sufficient that the cost objective decreases at every time step and we cannot hope to obtain a global optimum anyway (We can only approximate it with repeating the optimization procedure with multiple random seeds, which we do, see Algorithm 1).
>
>
> ## Issue 3
>
> We thank the reviewer for pointing out the different experimental settings and would like to further clarify the intent of the experiments. As correctly pointed out by the reviewer, we provide two “types” of comparisons.
>
> #### **Types of experimental comparisons**
>
> 1. One-shot standardized experiments in Section 3.1:
> Here we simply tested the effectiveness of _one_ compression step in conjunction with various amounts of retraining afterwards. The emphasis hereby is deploy the _exact_ same compress+retrain pipeline across all methods to test the effectiveness in a standardized environment. We believe this to be a valuable insight given that many different compression pipeline oftentimes deploy vastly different training schedules and hyperparameters configurations. Our one-shot experiments aim to “normalize” some of these differences by providing a standardized platform for comparison
>
>
> 2. Best effort benchmark comparisons in Section 3.2:
> We also agree with the reviewers that some of these methods (e.g. `L-Rank`) are designed with a specific train+compress schedule in mind. Hence, we provided comparisons in Section 3.2 to other papers using the respective “best effort” results reported in each of the papers.
>
> #### **Reasoning for experiments**
>
> We believe that these two distinct sets of experiments provide a much more thorough experimental validation of the different approaches compared to a single type of comparison. We were also very clear throughout Section 3 as well as in the appendix in explaining the different experimental setups to ensure that the experiments can be properly evaluated by a potential reader or reviewer.
>
> #### **`ALDS` outperforms `L-Rank`**
>
> Finally, we would like to ask the reviewer to clarify what they meant when they mentioned that the `L-Rank` comparison method performs _on par_ with our method (`ALDS`) in Table 2. Specifically, in Table 2 we report a *77.9%* and *81.4%* reduction in FLOPs (CR-F) at -0.21 and -0.41 difference in accuracy, respectively, compared to the original model. On the other hand, `L-Rank` reports 66.77% reduction in FLOPs for -0.13 difference in accuracy. Contrary to what the reviewer claims, this seems like a clear improvement over `L-Rank`.
>
> ## Issue 4
>
> #### **Reference to Theorem 1 in manuscript**
>
> Thank you for bringing up this issue. We would like to again point the reviewer to Section A.4, where we state in lines 778-781 that _we use the derived upper bound_ to compute the cost objective of Equation (8) and that we are _saving it in the lookup table._ For increased clarity we are also going to summarize the answer to your question below:
>
> #### **Relevance of Theorem 1 as stated in manuscript**
>
> We use Theorem 1 to quickly evaluate the objective function in Equation (8) during the optimization procedure. Specifically, we can express the relative error as a function of the rank and we thus only need to solve the underlying SVD for each layer once for each value of $k^\ell$. Without Theorem 1 we would need to compute the relative error (operator norm) for each pair $j^\ell, k^\ell$ separately. This would in turn result in a significant slowdown of the runtime of Algorithm 1 (practical runtimes are reported on lines 239, 240).
>
> ## Final Remarks
> Finally, we thank the reviewer for the comments that helped us further improve the quality of this paper. We hope that the reviewer will significantly increase the score as we have answered all of the concerns. We will be happy to answer more questions if there are any.

---

> > ### Comment · Reviewer_piFe · 2021-08-21
> > **Thanks for addressing issues 2,3, and 4**
> >
> > Thanks for addressing issues 2, 3, and 4. I believe readers would strongly benefit from having OptimalRanks and OptimalSubspaces being explicitly written and explained in the main paper (as you did in your answer), however, there might be no space for that, but a dedicated explanation in suppl.mat. will be sufficient. I would encourage authors to give direct links to suppl.mat from the main paper (for issues 2 and 4).
> >
> > Regarding your question in issue 3. I believe authors of L-rank released updated low-rank models available at the link [1] with better errors and flops counts. However, as far as I can tell from the page, this update was done recently, after the neurips submission deadline.
> >
> >
> > 1. https://github.com/UCMerced-ML/LC-model-compression/tree/master/examples/low_rank_alexnets

---

> > ### Author Response · Authors · 2021-08-21
> > **Thank you for Engaging with us - Issues 2, 3, 4**
> >
> > We thank the reviewer for this suggestion that helped us further improve our paper. We also really appreciate that the reviewer is engaging with us during this discussion period.
> >
> > ### **Issues 2,4**
> >
> > We will add full, detailed explanations in the supplementary material. As per the reviewer’s suggestion, we will provide direct links and reference the supplementary from the relevant sections in the main paper.
> >
> > Please let us know if you would like to obtain a preliminary version of the updated supplementary material. We would be happy to provide an anonymous download link at your request.
> >
> > ### **Issue 3**
> >
> > Thank you for the reference to the updated code repository.
> >
> > As the reviewer pointed out, the authors of `L-Rank` seemed to have updated their results after the NeurIPS submission. On the other hand, we reported the numbers from the official paper as accepted at CVPR 2020.
> >
> > We would be happy to incorporate their latest results as concurrent results into our comparisons in the next version of our manuscript.
> >
> > We also investigated how we could further compare to `L-Rank`.
> >
> > Unfortunately, there are no other ImageNet benchmarks available for `L-Rank` as they have only reported results for AlexNet on ImageNet.
> >
> > However, in order to ensure a better comparison to `L-Rank`, we now also compare our `ALDS` results with the results on the CIFAR10 dataset that the `L-Rank` paper reports. Specifically, `L-Rank` also reports results for ResNet20 and VGG16 on CIFAR10:
> >
> >
> > | **ResNet20, CIFAR10** | Accuracy | Acc. Delta | CR-P | CR-F |
> > | --- | --- | --- | --- | --- |
> > | Uncompressed | 91.39 | 0.00% | 0.0% | 0.0 % |
> > | |
> > | `L-Rank` | 91.24% | -0.15% | 35.1% | 35.1%
> > | `L-Rank` | 90.78% | -0.61% | 43.8% | 44.1%
> > | `L-Rank` | 90.51% | -0.88% | 57.6% | 57.1%
> > | `L-Rank` | 90.13% | -1.26% | 68.4% | 66.8%
> > | `L-Rank` | 89.03% | -2.36% | 76.1% | 73.9%
> > | `L-Rank` | 87.11% | -4.28% | 80.6% | 78.0%
> > | |
> > | `ALDS` | 91.48% | +0.09% | 64.58% | 55.95%
> > | `ALDS` | 90.92% | -0.47% | 74.91% | 67.86%
> > | `ALDS` | 90.71% | -0.68% | 79.01% | 71.59%
> > | `ALDS` | 89.51% | -1.88% | 87.68% | 83.23%
> > | `ALDS` | 88.80% | -2.59% | 89.65% | 85.23%
> >
> >
> > | **VGG16, CIFAR10** | Accuracy | Acc. Delta | CR-P | CR-F |
> > | --- | --- | --- | --- | --- |
> > | Uncompressed | 92.78% | 0.00% | 0.0% | 0.0 % |
> > |  |
> > | `L-Rank` | 93.32% | +0.53% | 88.7% | 81.5%
> > | `L-Rank` | 92.78% | +0.00% | 89.8% | 83.3%
> > | `L-Rank` | 92.72% | -0.06% | 91.2% | 85.5%
> > | `L-Rank` | 92.63% | -0.15% | 91.9% | 86.8%
> > |  |
> > | `ALDS` | 93.07% | +0.29% | 94.9% | 84.0%
> > | `ALDS` | 92.67% | -0.11% | 95.8% | 86.2%
> > | `ALDS` | 92.26% | -0.52% | 97.0% | 89.0%
> >
> > In all of these instances, we can observe that `ALDS` outperforms `L-Rank` both in terms of parameter and FLOP reduction. Note that a higher compression ratio (CR-P and CR-F for parameters and FLOPs, respectively) is better.
> >
> > Hopefully, this will convince the reviewer further that our method is very competitive with state-of-the-art methods.

---

> ### Author Response · Authors · 2021-08-28
> **Please inform us of any outstanding issues**
>
> We would like to thank Reviewer piFe one more time for already engaging with us during the discussion period. Your feedback is truly appreciated and has already helped significantly in improving the quality of our manuscript.
>
> As the discussion period is coming to an end, we would like to reach out one more time to inquire about potential further issues or questions you might have that we could address. If so, we will do our best in order to respond in a timely manner.
>
> From re-reading our conversation, we sincerely hope that we have addressed all of your concerns at this point and that you will consider raising your score and voting for acceptance of the paper.
>
> Specifically, we believe we were able to adequately address your initial concern about the differences between our method (ALDS) and SVD. In summary, ALDS constitutes _a generalization_ of SVD and thus is at least as good as SVD since the classic SVD is one possible outcome of the automatic decomposition procedure in ALDS.
>
> Please do not hesitate to reach out with any remaining concerns and we genuinely hope that our responses have convinced you to vote for the acceptance of our paper.
>
> Thank you again!

---

### Author Response · Authors · 2021-08-10
**General Response**

#### **Thank you**

We thank all the reviewers for their positive feedback and constructive criticism of our work. We greatly appreciate the time and effort by the reviewers to help us improve our work.

#### **Positive feedback**

Reviewer piFe mentions that _the paper is well written, and the method’s main idea is presented well_. Reviewer auNv highlights that _compared to the other [l]ow rank method[s] [we] show that the proposed ALDS has a better prediction performance at the considered compression ratio._
Moreover, Reviewer AgkJ mentions that _the paper is well organized and the method is easy to follow_; that _the proposed method is well justified theoretically_; and that _the results are promising._

#### **Our main contributions**

In summary, our main contribution is a novel approach to neural network compression via low-rank decomposition. Our algorithm hinges on the idea of compressing each convolutional (or fully-connected) layer by “slicing” its channels into multiple groups and decomposing each group via low-rank decomposition. This is the local step that we use to compress individual layers. We then leverage our layer-wise compression (and theoretical insights) to frame the compression problem as an optimization problem where we wish to minimize the maximum compression error across layers and propose an efficient algorithm towards a solution. By automatically assigning a per-layer compression ratio we can efficiently search over a large set of possible decompositions to find a more optimal compression. Our algorithm (ALDS) is backed up by a wide range of experimental evalutions where we outperform previous methods for low-rank compression.

#### **Reference to detailed responses**

We will be addressing each reviewer’s comments in detail below in separate responses. If there are any remaining concerns, please do not hesitate to raise these issues and we would be more than happy to address them in a timely manner.

#### **Further discussion and improved scores**

Overall, we believe we were able to address the reviewers’ concerns to an extent that will hopefully convince the reviewers to raise their scores.

We look forward to further engaging with you during the up-coming discussion period.

---

### Author Response · Authors · 2021-08-28
**Summary of Reviews and Discussions: Concerns Are Addressed**

Dear Reviewers, ACs, and SACs,

First off, we would like to express our deep gratitude for spending the time to review and assess our paper as well as providing us with insightful comments throughout the review process. Your feedback has already helped us to significantly improve our paper.

As the discussion period is coming to an end, we would like to take a moment to summarize the reviews, our responses, and the interactions with the reviewers we had up to this point.

We feel this is particularly important as so far we have only had the chance to partially interact with one reviewer (Reviewer piFE) during the discussion period to resolve lingering issues.

At the same time, we believe we were able to more than adequately address all the reviewers’ concerns and issues.

Consequently, we cannot identify any remaining shortcomings of our paper that would preclude it from acceptance or would justify the comparably low scores at this point in the review process.

Again, we would like to emphasize our appreciation for the openly communicated review process. We are here to address any lingering issues and comments at short notice as the review process is coming towards an end.

We will now proceed with summarizing each of the reviews and interactions below by adding individual comments for each review right below this post.

Thank you and we look forward to hearing back,

The `ALDS` authors

---

> ### Author Response · Authors · 2021-08-28
> **Summary: Reviewer piFe**
>
> The reviewer raised four main issues in their initial review.
>
> **Issues 2 and 4.**
> Two of them (Issue 2 and 4) were related to clarifying some details with regards to Algorithm 1 and Theorem 1. We have addressed them by pointing the reviewer to the respective paragraphs in the current version of the manuscript and the supplementary material. Moreover, where appropriate, we provided additional clarification and detailed explanations to ensure the material is fully accessible (see our response **[here](https://openreview.net/forum?id=BvJkwMhyInm&noteId=duyf8QGVybh)**).
>
> **Issue 1.**
> Another issue (Issue 1) pertains to an empirical analysis and comparison to SVD that the reviewer suggested. Specifically, it seems that their initial negative assessment of our paper stemmed in big parts from a misinterpretation of our method and a resulting unfair comparison in their proposed Matlab evaluation script.
>
> We have mostly addressed this issue by now and we are hoping the reviewer will reach out again to continue engaging with us.
>
> At this point, we would also like to emphasize that ALDS is **a generalization** of SVD to multiple subspaces (the parameter “$k$”). Naturally, therefore ALDS is at least as good as SVD ($k=1$) for an appropriately chosen $k$. Since $k$ is determined automatically according to Algorithm 1 in a way that minimizes the relative error, ALDS cannot be worse than SVD in practice. This is also confirmed empirically in our ablation study in Section 3.3.
>
> **Issue 3.**
> Finally, the reviewer was concerned about the comparison with L-Rank (Issue 3). Our method outperforms L-Rank on AlexNet (ImageNet) according to their official CVPR paper. But the authors of L-Rank seem to have recently posted improved results on their [GitHub repo](https://github.com/UCMerced-ML/LC-model-compression/tree/master/examples/low_rank_alexnets#update), from which we can conclude that our method performs _competitively_ with theirs in this particular scenario.
>
> Unfortunately, the L-Rank paper only reports ImageNet results on AlexNet, a fairly outdated architecture, but no other architecture. We, whereas, also report results on ImageNet for ResNet18 (see Section 3.2) and MobileNetV2 (see our response **[here](https://openreview.net/forum?id=BvJkwMhyInm&noteId=28Upw1tqCj)**). Moreover, we now also compare to the L-Rank CIFAR-10 results (see our response **[here](https://openreview.net/forum?id=BvJkwMhyInm&noteId=8CBFjcENix)**), where we consistently outperform L-Rank.
>
> Therefore, we believe it is reasonable to conclude that our method performs at least competitively or outperforms L-Rank and this particular comparison should not preclude our paper from acceptance.
>
> **Conclusion.**
> In summary, we hope we are able to highlight how we have not only significantly improved our manuscript thanks to the insightful comments made by the reviewer but also convinced the reviewing team that all of the concerns are being more than sufficiently addressed.

---

> ### Author Response · Authors · 2021-08-28
> **Summary: Reviewer auNv**
>
> In their initial review, Reviewer auNv identified four main points of confusion that they wanted to have addressed during the rebuttal period.
>
> We could answer a majority of those questions by pointing the reviewer to the appropriate sections in our paper, where the material was already clearly accessible.
>
> In fact, some of the concerns raised in the review (e.g. channel slicing and selection of $k$, see **[here](https://openreview.net/forum?id=BvJkwMhyInm&noteId=Dk81bBRiraE)**) are the very research question we are trying to investigate in this paper in the first place. Consequently, these questions are already abundantly addressed throughout the paper.
>
> However, we also tried our best during that process in identifying any shortcomings and missing details from our current version of the manuscript to ensure that the material is fully accessible to a broad audience.
>
> As discussed with Reviewer piFe as well, we will ensure that Algorithm 1 is much more accessible beyond the current explanation in the main paper and in the supplementary material. We are going to provide the detailed explanation we also provided in our responses to Reviewers piFe, auNv.
>
> From there, we could not identify any more open issues raised by the reviewer and again cannot identify further reasons for the (currently too) low score.
>
> Unfortunately, we were not able to interact with the reviewer so far but we remain hopeful that the reviewer will get back to us to discuss any other potential issues. We are looking forward to it!

---

> ### Author Response · Authors · 2021-08-28
> **Summary: Reviewer AgkJ**
>
> The reviewer identified three issues and concerns, all of which have been addressed very thoroughly as we feel.
>
> **First,** the reviewer requested additional ImageNet experiments on an efficient architecture, such as MobileNetV2. Since the initial review, we have actually run these experiments for multiple different compression ratios and comparison methods to highlight the effectiveness of our method even on such a less common benchmark. Our results can be found in our response **[here](https://openreview.net/forum?id=BvJkwMhyInm&noteId=28Upw1tqCj)**.
>
> In fact, we would like to highlight that to the best of our knowledge we are one of the first papers to conduct tensor decomposition experiments on an efficient architecture like MobileNet. We believe these results will significantly strengthen our paper while also addressing the reviewer’s concern.
>
> In comparison, the L-Rank paper that we thoroughly discussed with Reviewer piFe only reported compression results on AlexNet, a significantly overparameterized architecture, and the paper was just published last year at CVPR ’20.
>
> **Second,** the reviewer was concerned about our comparison being based on the parameter count. To address this issue, we pointed the reviewer to Section 3.2 as well as to our supplementary material, where we report both parameter reduction and FLOP reduction for all of our experiments as is common in the related work (cf. L-Rank).
>
> **Third,** there seems to have been a misunderstanding on the reviewer’s end that we are not allowed to use an author-year style for our citations if we understood the reviewer correctly. This is, however, clearly allowed at NeurIPS.
>
> **In summary,** the main improvement we needed to make was to add another experiment on ImageNet, which we did. We also want to emphasize that many related papers only report results on 1-2 ImageNet architectures such as we did in our original manuscript. Now we have 3. However, we were more than happy to conduct these additional experiments as they only strengthen our contribution.
>
> Again, we have been able to address all the concerns outlined by the review. We would love to have a chance to continue engaging with the reviewer. But as it stands now and given how we generously responded to the issues, we believe that our paper merits a higher score from the reviewer.

---

### Decision · Program_Chairs · 2021-09-27

**Decision:**

Accept (Poster)

**Comment:**

The paper suggests a network compression scheme using low-rank approximation of linear layers (either convolutional, or dense).  So far not much new here, except that this paper deals with the combinatorially hard problem of optimizing the matrix decomposition (compression) parameters in all layers simultaneously, so as to minimally sacrifice accuracy.  They suggest an EM-like strategy for solving this optimization problem.

There has been a considerable amount of discussion between the reviewers and the authors, including exchange of code snippets, which is rare.  After reading through this exchange, I became more inclined to accept the paper.  I also think that the optimization problem discussed by the paper does not get enough attention in the network compression literature, and the approach the authors suggest is very reasonable.  Therefore, I choose to recommend acceptance in spite of the average “borderline/reject” score average.